# Tracing the role of Arctic shelf processes in Si and N cycling and export through the Fram Strait: Insights from combined silicon and nitrate isotopes

Margot C.F. Debyser[1], Laetitia Pichevin[1], Robyn E. Tuerena[2], Paul A. Dodd[3], Antonia Doncila[1], Raja S. Ganeshram[1]

[1]School of Geosciences, University of Edinburgh, Edinburgh, EH9 3FE, United Kingdom
[2]Scottish Association for Marine Science, Dunstaffnage, PA37 1QA, United Kingdom
[3]Norwegian Polar Institute, Tromsø, 9296, Norway

*Correspondence to*: Margot C.F. Debyser (margot.debyser@ed.ac.uk)

**Abstract.** Nutrient cycles in the Arctic ocean are being altered by changing hydrography, increasing riverine inputs, glacial melt and sea-ice loss due to climate change. In this study, combined isotopic measurements of dissolved nitrate ($\delta^{15}$N-NO$_3$ and $\delta^{18}$O-NO$_3$) and silicic acid ($\delta^{30}$Si(OH)$_4$) are used to understand the pathways that major nutrients follow through the Arctic ocean. Atlantic waters were found to be isotopically lighter ($\delta^{30}$Si(OH)$_4$= +1.74‰) than their polar counterpart ($\delta^{30}$Si(OH)$_4$= +1.85‰) owing to partial biological utilisation of dissolved Si (DSi) within the Arctic ocean. Coupled partial benthic denitrification and nitrification on Eurasian Arctic shelves leads to the enrichment of $\delta^{15}$N-NO$_3$ and lighter $\delta^{18}$O-NO$_3$ in the polar surface waters ($\delta^{15}$N-NO$_3$ = 5.44‰, $\delta^{18}$O-NO$_3$ = 1.22‰) relative to Atlantic waters ($\delta^{15}$N-NO$_3$ = 5.18‰, $\delta^{18}$O-NO$_3$ = 2.33‰). Using a pan-Arctic DSi isotope dataset we find that the input of isotopically light $\delta^{30}$Si(OH)$_4$ by Arctic rivers and the subsequent partial biological uptake and biogenic Si burial on Eurasian shelves are the key processes that generate the enriched isotopic signatures of DSi exported through Fram Strait. A similar analysis of $\delta^{15}$N-NO$_3$ highlights the role of N-limitation due to denitrification losses on Arctic shelves in generating the excess dissolved silicon exported through Fram Strait. We estimate that around 40% of DSi exported in Polar Surface Waters through Fram Strait is of riverine origin. As the Arctic ocean is broadly N-limited and riverine sources of DSi are increasing faster than nitrogen inputs, a larger silicic acid export through the Fram Strait is expected in the future. Arctic riverine inputs therefore have the potential to modify the North Atlantic DSi budget and are expected to become more important than variable Pacific and glacial DSi sources over the coming decades.

## 1 Introduction

The dissolved macronutrients nitrate (NO$_3^-$) and silicic acid (Si(OH)$_4$) are key nutrients in sustaining marine primary production in the Arctic ocean, and have distinct sources from the Atlantic and Pacific Oceans (Tremblay et al., 2015). Additionally, river and coastal erosion contribute dissolved silicon (DSi) and nitrate which fuel approximately 30% of Arctic-wide net primary productivity (Terhaar et al., 2021). The Greenland Ice Sheet has also been suggested as an important source

of DSi to the Arctic ocean (Hatton et al., 2019; Hawkings et al., 2017). It has been estimated that >85% of DSi from riverine sources is not consumed by phytoplankton (Le Fouest et al., 2013) and is exported out instead, but the controlling processes of this remain unclear. Thus, an integrated understanding of the relative importance of sources to the internal cycling of DSi and the controls on the export of DSi to the North Atlantic is lacking. Stable isotope measurements of nitrate ($\delta^{15}$N-NO$_3$ & $\delta^{18}$O-NO$_3$) and dissolved silicon ($\delta^{30}$Si(OH)$_4$) can provide useful insights into nutrient sources and cycling within the ocean

(Brzezinski et al., 2021; Sigman et al., 2000; Varela et al., 2016), particularly when both isotopes are combined (Grasse et al., 2016; De Souza et al., 2012). In this study, we present the first full profiles of $\delta^{30}$Si(OH)$_4$ measurements in Fram Strait and over the East Greenland shelf in conjunction with nitrate isotopes to examine the controls on DSi export through the Fram Strait and suggest potential future scenarios.

In the Arctic ocean, primary production is controlled by complex interactions between light availability and nutrient limitation (Giesbrecht and Varela, 2021; Popova et al., 2012; Yool et al., 2015) which are highly variable both spatially and temporally. Nitrogen is the primary limiting nutrient for primary production in the Eurasian Arctic (Krisch et al., 2020; Tuerena et al., 2021a) and sedimentary denitrification on shallow Arctic shelves play an important role in limiting nitrogen availability (Fripiat et al. 2018; Granger et al. 2018) making the Arctic ocean a net sink of nitrate overall (Yamamoto-Kawai et al., 2006).

In contrast, there is an excess of DSi in the Arctic ocean and a disproportionally large amount of DSi is exported to the North Atlantic via Fram Strait and the Canadian Arctic Archipelago. Budget estimates have shown that the Arctic ocean contributes to more than 10% of the DSi entering the North Atlantic (Torres-Valdés et al., 2013).

The excess of DSi in the Arctic Ocean's DSi budget is attributed to Pacific water, which enters the Arctic through the Bering

Strait, but also freshwater sources, as highlighted in Figure 1. The Arctic ocean receives a disproportionally large volume of freshwater relative to its area (>10% of the world's riverine discharge) from several of the world's largest rivers, such as the Ob, Yenisei, Lena and Kolyma rivers which discharge onto the Eurasian shelves. These four rivers alone provide 1755 km$^3$ of freshwater to Arctic shelves annually, along with 135 x 10$^9$ g of nitrate and 4816 x 10$^9$ g of DSi (Holmes et al., 2012), which fuels coastal and Arctic-wide productivity, subsequently transported through the Transpolar Drift (TPD) (Charette et al., 2020;

Terhaar et al., 2021). Arctic glacial meltwaters provide a potentially significant contribution to the Arctic's nutrient budget (Hatton et al., 2019; Meire et al., 2016), with DSi and amorphous Si inputs from the Greenland Ice sheet estimated to constitute around 37% of riverine fluxes in the coastal regions of Arctic Seas (Hawkings et al., 2017). However, large quantities of DSi are removed within fjords (Hopwood et al., 2020) and the fraction that is exported from Greenland and Svalbard fjords into the open ocean remains poorly documented.


Atlantification is leading to changes in sea-ice cover and stratification of the Eurasian basin (Arthun et al., 2012; Lind et al., 2018) and increasing nutrient availability in the surface ocean (Randelhoff et al., 2018; Tuerena et al., 2021a). Meanwhile, DSi concentrations from Atlantic Waters (AW) are decreasing in the sub-Arctic regions (Hátún et al., 2017) and the inflow of

Pacific water is increasing (Woodgate, 2018). Riverine freshwater inputs have been increasing in the Eurasian sector

(Mcclelland et al., 2006) and nutrient fluxes are increasing in rivers with degrading permafrost (Frey et al., 2007; Frey and McClelland, 2009; Zhang et al., 2021). All of these changes have widespread impact on phytoplankton dynamics (Ardyna and Arrigo, 2020). In response, the nutrient budgets of the Arctic ocean are expected to change, with potential repercussions on downstream ecosystems and Atlantic nutrient budgets. In order to predict such impacts, a better understanding of the relative importance of Arctic nutrient sources and internal cycling is needed.


Fram Strait is both an inflow and outflow gateway and a key area of exchange between the Arctic and the North Atlantic. On the Eastern side, warm, saline AW originating from the subpolar and subtropical gyre of the North Atlantic flows northward in the surface intensified West Spitsbergen Current. On the Western side, Polar Surface Water (PSW) carries cold, fresh Arctic-originating water and sea-ice into the Subpolar North Atlantic Ocean in the upper (ca. 250m) part of the water column (Dodd

et al., 2012; Rudels et al., 2002; de Steur et al., 2009). PSW is relatively low in nitrate, carrying the signal of Pacific nutrient stroichiometry and benthic denitrification to the Atlantic Ocean through low N:P ratio (Dodd et al., 2012). In contrast to PSW, AW has relatively high nitrate concentrations but is poor in DSi ($\cong 5\mu M$) as this key nutrient is depleted in the Atlantic through uptake by silicifying phytoplankton species during its northward movement. The stoichiometry of DSi availability compared to nitrate (DSi:N <1) in AW in Fram Strait suggests phytoplankton blooms experience DSi limitation prior to nitrate limitation

(Krause et al., 2018, 2019; Tuerena et al., 2021a).

Nitrate removal processes within the Arctic ocean are reflected in the nitrate isotopic signatures of $5.5 \pm 0.4$‰ for $\delta^{15}N\text{-}NO_3$ and $1.3 \pm 0.4$‰ for $\delta^{18}O\text{-}NO_3$ of PSW (Tuerena et al., 2021a), which are significantly different from incoming AW signatures of $5.1 \pm 0.2$‰ for $\delta^{15}N\text{-}NO_3$ and $2.4 \pm 0.3$‰ for $\delta^{18}O\text{-}NO_3$. The difference between these two water masses reflects benthic

denitrification on shallow Eurasian shelves, also termed coupled partial nitrification-denitrification (CPND), which increases $\delta^{15}N\text{-}NO_3$ while decreasing $\delta^{18}O\text{-}NO_3$ producing an associated increase in the parameter $\Delta(15\text{-}18)$, defined as $\delta^{15}N\text{-}NO_3 - \delta^{18}O\text{-}NO_3$, through the release of isotopically heavy ammonia from sediments (Fripiat et al., 2018; Granger et al., 2018). PSW transports high DSi concentrations from Pacific and riverine influence: $\delta^{30}Si(OH)_4$ of Pacific water is $\cong +1.4 \pm 0.2$‰ (Reynolds et al. 2006) and the isotopically light source of DSi is traced through the Bering strait and into the upper halocline waters of

the Arctic Ocean (Giesbrecht et al., 2022). Pacific $\delta^{30}Si(OH)_4$ is lighter than North Atlantic signatures ($\delta^{30}Si(OH)_4 \geq +1.7$‰) which are enriched to a greater extent from the Southern Ocean source signal as DSi is depleted through partial uptake and subsequent burial of DSi in the North Atlantic (Brzezinski and Jones, 2015; De Souza et al., 2012). Siberian rivers have high seasonal and regional variability in their isotopic signatures, which are isotopically light from weathering processes in Arctic rivers, leading to fractionation of the isotope from the local bedrock (Pokrovsky et al., 2013; Sun et al., 2018). The mean

$\delta^{30}Si(OH)_4$ of DSi from major rivers to the Arctic Ocean is estimated at $+1.3 \pm 0.3$‰ (Sun et al., 2018). In addition to these

sources, benthic supply of DSi to Arctic shelves was recently documented in the Barents Sea which adds light isotopes to shelf waters (Ward et al., 2022a, 2022b), although the magnitude of this flux on a basin-wide scale is currently unknown.

While Arctic sources of DSi are isotopically light, Arctic polar surface waters are isotopically heavy ($\delta^{30}$Si(OH)$_4$ $\cong$ +1.8 ± 0.1‰) with isotopically heavy deep basins (Brzezinski et al., 2021; Giesbrecht et al., 2022; Varela et al., 2016). This heavy isotopic enrichment is attributed to physical processes (Liguori et al., 2020) and biological modification within surface waters (Giesbrecht et al., 2022; Varela et al., 2016). A recent study however highlights the importance of biological productivity and biogenic Si burial of riverine DSi in generating these enriched Arctic signatures (Brzezinski et al., 2021). Although isotopic signatures have been measured up to 60˚N in the Atlantic Ocean (De Souza et al., 2012; Sutton et al., 2018), no direct measurements of $\delta^{30}$Si(OH)$_4$ are available from Atlantic-Arctic Gateways such as the Fram Strait. Therefore, $\delta^{30}$Si(OH)$_4$ signatures of modified inflowing AW in the Arctic ocean and outflowing PSW and the contributions from East Greenland shelves are unknown.

This study fills this crucial gap in the Arctic silicon isoscape, documenting isotope signatures and nutrient cycling processes in Fram Strait, focussing on the upper water masses. We use a combination of geochemical parameters ($\delta^{30}$Si(OH)$_4$, $\delta^{15}$N-NO$_3$, $\delta^{18}$O-NO$_3$, $\Delta$(15-18), N* & Si*) alongside hydrographic data (salinity, temperature, mixed layer depth) to explore the sources and internal cycling of DSi in the water masses exported through the Fram Strait. We then proceed to put these in the context of pan-Arctic isotope datasets and evaluate the implications of Arctic nutrient cycling on how nutrient export is likely to change in the future with ongoing climate change.

## 2 Method

### 2.1 Sample collection

Samples were collected from two CTD sections across the Fram Strait (JR17005 & FS2018) and from CTD profiles near the Ile-de-France between 2017-2019 (Table 1). The CTD package was equipped with a SBE911plus CTD system recording multiple parameters (conductivity, temperature, pressure & salinity). Salinity was calibrated on-board using an Autosal 8400B salinometer (JR17005) and a Guildline Portasal salinometer (FS2017-2019). Samples for dissolved inorganic nutrient analysis were collected from Niskin bottles and stored in pre-cleaned HDPE bottles which were frozen at -20 °C immediately after collection. Samples for isotopic analysis were filtered inline using Nuclepor polycarbonate membranes (0.4μM porosity) into acid-cleaned polypropylene bottles and stored at -20˚C (nitrate isotopes) or acidified at 0.1% v/v with 12M HCl and stored at 4°C (silicon isotopes).

## 2.2 Dissolved inorganic nutrient measurements

Dissolved inorganic nutrient concentrations for JR17005 were determined from frozen samples on autoanalysers following standard colorimetric methods on a Bran and Luebbe QuAAtro 5-channel autoanalyser at the National Oceanographic Centre UK (Brand et al., 2020). Detection limit for nutrient analysis was 0.1μM and 0.03μM for DSi and nitrate respectively with accuracy with respect to CRMS of 2.75% and 0.91% (Brand et al., 2020). For FS2018, nutrients were analysed following methods from Hansen and Koroleff, (1999) & Schnetger and Lehners, (2014) on a SmartChem 200 discrete analyser at the Technical University of Denmark (FS2017-19) and calibrated using OSIL nutrient standards. Analytical precision is of 2% and the detection limit was of 0.4μM for nitrate and 0.1μM for DSi. While measurement from frozen is suboptimal for silicic acid concentrations, separate non-frozen samples could not be collected for nutrients due to sampling and shipping restrictions. DSi concentrations were independently checked at the University of Edinburgh from the silicon isotope samples (acid preserved) during analysis with the HACH reagent method. Both datasets from frozen and acidified were in very good agreement and frozen samples were not found to have lower DSi concentrations. DSi concentrations from FS2018 also closely align with concentrations measured in the same water masses in JR17005 below the seasonal layer in the upper 500m of the water column, and align with published concentrations in the literature.

## 2.3 Nitrate isotope analysis

$\delta^{15}$N-NO$_3$ & $\delta^{18}$O-NO$_3$ were measured using the bacterial strain of *P. aureofaciens* following the denitrifier method (Casciotti et al., 2002; Sigman et al., 2001). Measurements were corrected using international reference standards IAEA-N3 and USGS-34 in each run, as well as an internal standard of North Atlantic Deep Water ($\delta^{15}$N-NO$_3$ = 4.92 ± 0.12 ‰, $\delta^{18}$O-NO$_3$ = 1.88 ± 0.45‰) for inter-run comparability, with standard reproducibility across runs of ±0.1‰ and ±0.4‰ for $\delta^{15}$N-NO$_3$ & $\delta^{18}$O-NO$_3$ respectively. Final values were corrected using the correction scheme described in Weigand et al. (2016) and following Tuerena et al. (2021a, 2021b) for inter-comparability of datasets in the Atlantic-Arctic region.

## 2.4 Silicon isotope analysis

DSi concentrations are very low in the Arctic ocean (<10μM), as such, previous protocols from Brzezinski et al. (2003) and Reynolds et al. (2006), originally based on the Magnesium Induced Coprecipitation (MAGIC) method described in Karl and Tien (1992), were adapted to allow measurements at concentrations below 10μM. DSi of seawater (40ml of sample) was co-precipitated in two steps along with brucite using 1M NaOH. Precipitate was recovered after 24h by centrifugation, re-dissolved using 6M HCl and diluted to 2ppm Si. The solution was further purified by loading 0.5ml of the solution onto pre-cleaned 1.8ml Biorad AG50W-X8 cation-exchange resin columns.

The isotopic composition of the prepared solution was determined by MC-ICP-MS on a Nu Plasma II instrument at the University of Edinburgh using standard-sample bracketing and calculated from the permil deviation from isotopic reference material NBS28 (Georg et al., 2006), calculated as:

$$\delta\ ^{x}Si = \left( \frac{(\frac{^{x}Si}{^{28}Si})_{sample}}{(\frac{^{x}Si}{^{28}Si})_{NBS28}} - 1 \right) \times 1000[\permil]$$

Where $\delta^{x}Si$ is either $\delta^{29}Si$ or $\delta^{30}Si$. As per Fripiat et al. (2011a, 2011b) and Liguori et al. (2021), $\delta^{29}Si$ were converted to $\delta^{30}Si$ to improve reliability and global comparability of datasets (Cardinal et al., 2003, 2005), using the theoretical conversion factor of 1.96, calculated from the kinetic fractionation law (Young et al., 2002). The method of analysis and interferences are discussed in further detail in Supplementary Material S1.

Inter-run comparability & method reproducibility of measurements was checked with the international solid standard Big Batch & both high and low concentration seawater standards $Aloha_{1000}$ & $Aloha_{300}$. Average standard measurements for the period of this study is $Aloha_{1000}$= +0.67 ± 0.03‰, +1.32 ± 0.06‰ (n=16), BigBatch = -5.33 ± 0.02‰, -10.50 ± 0.04‰ (n=7) for $\delta^{29}Si(OH)_4$ and $\delta^{30}Si(OH)_4$ respectively (uncertainties of 1SD). Long-term reproducibility of converted $\delta^{30}Si(OH)_4$ is BigBatch = -10.49 ± 0.09‰ (n=58), $Aloha_{1000}$= +1.29 ± 0.08‰ (n=58) and $Aloha_{300}$= +1.70 ± 0.05‰ (n=30) compared to inter-laboratory measurements of BigBatch = -10.48 ± 0.2‰, $Aloha_{1000}$= +1.25 ± 0.2‰, $Aloha_{300}$= +1.66± 0.35‰ (Grasse et al., 2017; Reynolds et al., 2007). The reproducibility of the full chemical and analytical procedure, including chemical preparation and analytical measurements in separate MC-ICP-MS sessions, was additionally estimated on a subset of duplicate seawater samples ($n = 8$). The mean absolute difference between duplicate samples analysed was ± 0.04‰ (1 SD).

## 2.5 Derived parameters

Mixed-Layer Depth (MLD) is identified as the maximum depth at which the potential density was within $0.1 kgm^{-3}$ of the shallowest measurement (Peralta-Ferriz and Woodgate, 2015). MLD governs the depth for which nutrients resupply surface waters and to which planktons are mixed (Yool et al., 2015). In this study, PSW is defined as potential temperature ($\theta$) <0 °C and potential density ($\sigma_\theta$) <27.7 $kgm^{-3}$, and AW is defined as $\theta$>2°C and 27.7<$\sigma_\theta$<27.97 $kgm^{-3}$ or $\sigma_\theta$ <27.7 $kgm^{-3}$ and salinity >34.92 psu, as per Richter, Von Appen and Wekerle (2018).

The semi-conservative tracers N* & Si* were calculated from inorganic nutrient concentrations where N* = $NO_x$ – $PO_4^-$ x 16, adapted from Gruber & Sarmiento (1997), and Si* = DSi – $NO_x$ (Sarmiento et al., 2004). Both tracers are indicative of nutrient deviation from typical Redfield ratio, and highlight additional sources or processes through which nutrient become deficit (i.e Negative N* shows nitrate deficit in comparison to phosphate). The isotopic parameter $\Delta$(15-18) is calculated as $\Delta$(15-18) = $\delta^{15}N$-$NO_3$ - $\delta^{18}O$-$NO_3$. $\Delta$(15-18) captures variation in both isotopes, tracing sources and modification of nitrate (Rafter et al., 2013).

## 3 Results

### 3.1 Hydrography & mixed layer depth

Figure 2 shows temperature and salinity across Fram Strait in July-August 2018. The hydrographic situation is typical of the late summer season. Warm inflowing and recirculating sub-tropical originating AW is found primarily between 2.5W and the Eastern end of the section at 10°E, in the upper 500 m. Its core flows northward within the West Spitsbergen Current at 6-8˚E. Over the East Greenland Shelf, PSW dominates the upper 150m, extending from the western end of the section to the AW/PSW interface at about 3˚W. Re-circulating Atlantic Waters (RAW), underlays PSW on the East Greenland shelf, while Arctic Atlantic Water (AAW) is also found below AW at the foot of the East Greenland Shelf. We refer the reader to Rudels et al. (2002) for an overview of the properties of water masses found in Fram Strait.

The MLD did not exceed 100m for FS2018. Late-season MLD was deeper in AW than in PSW, occurring between 30-60m. MLD is significantly shallower over the East Greenland shelf, occurring at 5-10m. Hydrography and nutrient distribution of JR17005 follow roughly similar patterns as FS2018 apart from seasonal variations, and are previously described in Tuerena et al. (2021).

### 3.2 Nutrient concentrations

Panels a and b of Figure 3 show the nitrate and DSi concentrations in the upper 400m of the water column along the late-summer 2018 Fram Strait section. P-values reported are for t-tests between AW and PSW water masses. Nitrate concentrations were low across the section in the upper 50m of the water column from phytoplankton utilisation and dilution by low-nitrate freshwater sources. Below the mixed-layer depth, $NO_x$ is higher in AW ($12.10 \pm 0.98$ µM) than in PSW ($8.08 \pm 2.19$ µM, $p$ <0.01), consistent with export of low nitrate waters from the Central Arctic.

Negative N* reflecting a deficit of nitrate are evident on the East Greenland Shelf in panel c of Figure 3. In constrast, N* reaches near positive values in AW with an average of $-0.55 \pm 0.38$ below the MLD (Table 2). This highlights that nitrate is more depleted in PSW relative to dissolved phosphate, becoming potentially limiting to primary production towards the end of the biological growth season as nitrate concentrations approach 0µM.

DSi concentrations were low across the section above the MLD with stronger depletion at shallower depths and further West (Figure 3b). Comparison of DSi concentrations measured during May-June 2018 and August-September 2018  (Table 2) reveals that DSi concentrations were similarly higher in PSW ($4.28 \pm 2.93$ µM) and in AW ($3.19 \pm 1.20$ µM) in the mixed layer at the start of the season, and fell to similarly depleted concentrations by the end of the summer ($1.03 \pm 0.98$ µM and $1.26 \pm 1.11$ µM for AW and PSW respectively).

Below the mixed layer, DSi is low in AW ($5.42 \pm 0.71$ µM) from DSi poor Atlantic waters of sub-polar origins. DSi in PSW is higher than in AW ($6.64 \pm 1.71$ µM, $p$ <0.01), potentially reflecting Arctic sources of DSi to PSW. In the deep Fram Strait,

DSi concentrations vary locally, but generally increase with depth up to a concentration of $9.45 \pm 2.48\ \mu M$ in deep waters (Figure 4). On Figure 3d, strongly negative Si* in AW reflect the strong DSi deficit relative to nitrate in Atlantic-originating waters, while Si* closer to phytoplankton requirement in PSW illustrate excess DSi in PSW.

## 3.3 Isotopic signatures

Measured profiles of $\delta^{30}Si(OH)_4$ across Fram Strait are shown in panel h of Figure 3 (late summer data, for spring data see Supplementary Material S2) and Figure 4. Positive signatures of $\delta^{30}Si(OH)_4$ were measured throughout the water column, ranging from +1.34‰ to +3.16‰ for the entire section (Figure 3, panel h). The heaviest $\delta^{30}Si(OH)_4$ signatures were measured in the upper 100m of the section (Figure 4), consistent with fractionation from diatom uptake during growth. Below the MLD, mean $\delta^{30}Si(OH)_4$ for AW was $+1.74 \pm 0.06$‰ (Table 2), which aligns closely with measurements of waters from North

Atlantic origin (Brzezinski and Jones, 2015; De Souza et al., 2012). Conversely, DSi in PSW was isotopically heavier than DSi in AW ($p < 0.02$), the mean $\delta^{30}Si(OH)_4$ value for PSW was $+1.85 \pm 0.09$‰. This is comparable to measurements of the upper halocline layer in the Canadian basin ($\delta^{30}Si(OH)_4 = +1.84$‰) from Varela et al. (2016), and outflowing surface measurements from Brzezinski et al. (2021) in the TPD where $\delta^{30}Si(OH)_4 = +1.92$‰, and aligns with the heavy signatures of Arctic-originating waters in the North Atlantic (De Souza et al., 2012; Sutton et al., 2018).

In the deep waters of Fram Strait, $\delta^{30}Si(OH)_4$ are lighter than PSW (Figure 4), aligning with the gradient decrease of -0.15‰ over the full depth profile reported in Brzezinski et al. (2021). The measured signatures also align with measurements in the North Atlantic of Nordic originating endmembers (DW-$\delta^{30}Si(OH)_4. = +1.65 \pm 0.13$‰ & DSOW-$\delta^{30}Si(OH)_4. = +1.75 \pm 0.08$‰, De Souza et al., 2012). Light $\delta^{30}Si(OH)_4$ values were measured at the sediment interface of deep basins (Figure 4), showing potential interaction of benthic efflux of DSi from isotopically light pore-waters (Ehlert et al., 2016; Ward et al., 2022a, 2022b),

and remineralisation of isotopically lighter biogenic Si in the deep. This is also observed in Brzezinski et al. (2021) and Liguori et al. (2020) who found isotopically light measurements in the deep Nansen & Amundsen basins. Low sampling resolution within our dataset and the strong influence of local circulation precludes quantification of such local recycling processes from advective signals in Fram Strait with certainty.

    Figure 5 displays the full water column profiles of nitrate isotopes measured along the spring (JR17005) and late-summer

(FS2018) sections. $\delta^{15}N-NO_3$ is enriched in PSW (5.44‰) compared to AW (5.18‰, $p < 0.01$) while the $\delta^{18}O-NO_3$ is lighter in PSW (1.22‰) than in AW (2.33‰, $p < 0.01$, Table 2) following trends identified in Tuerena et al. (2021a). Panel g of Figure 3 illustrates the decoupling of both isotopes reflected in diverging $\Delta(15\text{-}18)$, indicative of CPND. A high confidence in accuracy and reproducibility of nitrate isotopes measurements in this study is obtained as the dataset aligns and follow the same trends as profiles reported in Tuerena et al. (2021a).

In surface waters, $\delta^{15}N-NO_3$ increase with reducing nitrate concentrations, which is consistent with biological uptake (Figure 5). This is also observed in most $\delta^{18}O-NO_3$ profiles apart from PSW profiles measured far onto the East Greenland shelf in

FS2018 (Figure 3), where remote signals of denitrification dominates over biological uptake signals, even in the upper water column. As shown in Figure 3, and summarised in Table 2, isotopic signatures of both dissolved silicon and nitrate isotopes closely follow the hydrography of water masses in spring and summer. In Fram Strait, a key area of exchange with the North

Atlantic where inflowing & outflowing water masses show strong differences in their physical properties, dissolved nitrate & silicon isotopes measurements provide insights into nutrient sources and cycling within the Arctic ocean and the pathways through which nutrient modification and exchange occurs.

## 4 Discussion

### 4.1 Using nutrient isotopes to examine Arctic nutrient cycling

**4.1.1 Trends between $\delta^{15}N$-$NO_3$, $\delta^{30}Si(OH)_4$ and nutrient utilisation**

In this section we compare $\delta^{15}N$-$NO_3$ and $\delta^{30}Si(OH)_4$ measurements with the fraction of nitrate and DSi remaining (f) in PSW and AW in Fram Strait (Figure 6, panels a and b). We define f as the fraction of nutrient remaining in the surface layer relative to concentrations below the MLD (table 2). f=1 indicates no nitrate or DSi has been used, f=0 indicates complete depletion of the nutrient inventory.

The fractionation of nitrate and DSi during phytoplankton uptake can be modelled by Rayleigh systematics (Altabet and Francois, 2001; Mariotti et al., 1981), and is often linked to local hydrography. Rayleigh systematics assume a closed system: i.e there is no import/export of the nutrient from the euphotic zone while it is being utilised by phytoplankton. In late spring and summer, the PSW layer in Fram Strait is largely a closed system as it is highly stratified. Nitrate and DSi are mainly replenished during winter destratification (Altabet and Francois, 2001). In this environment, $\delta^{15}N$-$NO_3$ and $\delta^{30}Si(OH)_4$ can be

expected to fall on a trend based on their isotopic effect (~2-6‰ for nitrate and ~1‰ for DSi globally), and are described by exponential trendlines in the Rayleigh field on Figure 6 (Varela et al., 2004). In areas of upwelling, or in a case where the resupply of nutrients to the euphotic zone occurs due to multiple stratification and destratification events throughout growth season, conditions are better modelled as open system, described by a linear trendline in the Rayleigh field. However, in low-nutrient zones such as the PSW layer in Fram Strait, nutrient uptake stoichiometry can be dictated by nutrient-limitation itself

rather than by the physical re-supply of nutrients (Hutchins and Bruland, 1998; Moore et al., 2013), which in turn can lead to a shift from open to closed system dynamics as the source of nutrients switches from new to remineralised.

During the growth season of 2018, nitrate in AW and PSW enriched in $\delta^{15}N$-$NO_3$ at lower nitrate concentrations, consistent with fractionation associated with nutrient uptake by phytoplankton (Figure 6)., AW follows the traditional isotopic effect of 5‰ and PSW follows the particularly low isotopic effect of 2‰. Nitrate fractionation in AW behaves between closed and open

system kinetics, with a shift from more closed conditions in spring towards more open conditions in summer. This corroborates with the relatively weak stratification of AW (Rudels et al., 2005), facilitating re-supply of nitrate and other nutrients over the

spring and summer growth season through destratification events such as those described by Tuerena et al. (2021a). In PSW, $\delta^{15}$N-NO$_3$ fractionation follows an exponential trend and behaves as a closed system in spring, indicative of the strong salinity stratification of PSW. A shift towards a mostly linear trend in summer is observed (Tuerena et al., 2021a), suggesting open system kinetics below the mixed layer. While a shift from open to closed system could be expected due to strengthening of stratification over the summer season, we observe a shift from closed to open systems instead. This is unlikely to reflect a change in hydrographic conditions, but indicates a shift towards consumption of regenerated nitrogen in nitrate depleted waters instead, thereby lowering ambient $\delta^{15}$N-NO$_3$ from expected trends. This is further supported by the equivalent trends observed in $\delta^{15}$N-PN (Figure S3, supplementary material S3).

The relationship between apparent remaining DSi and $\delta^{30}$Si(OH)$_4$ in AW follows closed system kinetics during the growth season (Figure 6) after DSi was drawn down to 1μM in AW in summer 2018 (Figure 4). DSi is strongly depleted in surface AW in summer, preventing direct $\delta^{30}$Si(OH)$_4$ measurements within the MLD and observation of shifts within the isotopic system. Our observations remain consistent with other studies which find DSi that DSi is one of the limiting nutrients to diatom growth in AW in the Eastern Fram Strait along with Fe (Krisch et al., 2020). In contrast to nitrate, biogenic Si is recycled less within the upper water column, preventing a switch to recycled nutrient sources later in the seasons. As DSi becomes fully utilised and ambient conditions become unfavourable to diatom growth, a shift towards non-siliceous species is expected in late summer along with a shift towards open system kinetics.

$\delta^{30}$Si(OH)$_4$ in PSW does not show a good fit with either of the fractionation models and measurements from within the MLD aren't consistent with fractionation associated with nutrient uptake by phytoplankton at lower nutrient concentrations. Trends of $\delta^{30}$Si-PSi are also inconsistent with any model (Figure S3, Supplementary Material S3). This suggests that unlike nitrate, DSi in PSW is not primarily controlled by biological processes, and its variations are more likely to be driven by physical mixing and dilutive effects instead. The decoupling of N & DSi isotopic systems is indicative that N is strongly limiting in the highly stratified PSW and prevents extensive DSi utilisation locally.

Nutrient uptake in surface AW is constrained by low DSi concentrations in limiting conditions for diatom growth, while uptake in PSW is constrained by strong nitrate limitation and DSi is only partially utilised. This indicates that the extent to which DSi is taken up is regulated by nitrate availability in PSW at Fram Strait. $\delta^{15}$N-NO$_3$ and $\delta^{30}$Si(OH)$_4$ show a strong link between the silicon and nitrogen cycles in Fram Strait as they regulate each other through availability, contributing to the asymmetry observed in nutrient exports across the strait (Torres-Valdés et al., 2013).

### 4.1.2 Upstream transformation of nitrate and DSi in PSW and AW in Fram Strait

The nutrient composition of polar waters exported through Fram Strait reflect their nutrient cycling history within the Arctic Ocean through altered DSi:N ratio and isotopic signatures. Figure 7a shows trends for $\delta^{15}$N-NO$_3$ vs $\delta^{18}$O-NO$_3$, displaying that fractionation due to uptake by phytoplankton assimilation follows the established fractionation ratio of 1:1 (Granger et al.,

2004; Sigman et al., 2005) but on separate fractionation lines. $\delta^{15}$N-NO$_3$ and $\delta^{18}$O-NO$_3$ of PSW plots on a fractionation line consistent with isotopically lighter sources of $\delta^{18}$O-NO$_3$, while $\delta^{15}$N-NO$_3$ and $\delta^{18}$O-NO$_3$ measured in surface AW follow a line consistent with isotopically heavier sources, suggesting different nutrient sources in AW and PSW.

The modification of nitrate in the Arctic ocean is readily apparent when plotting N* against $\delta^{18}$O-NO$_3$ (Figure 7b); as salinity decreases and the influence of Polar-originating waters increases, N* decreases, indicating a nitrogen deficit in relation to phosphate in PSW. Water samples with lower N* are accompanied by lighter $\delta^{18}$O-NO$_3$ signatures. This relationship is attributed to CPND in the Arctic ocean (Granger et al., 2011): settling particulate organic nitrogen from coastal productivity degrades at the sediment interface of the extensive shallow shelves and produces large sources of sedimentary ammonium. In shelves where sedimentary denitrification preferentially consumes isotopically light NO$_3$, the NH$_4^+$ thus released into the water column is isotopically heavy in $\delta^{15}$N. Subsequently, during nitrification, this benthic efflux of isotopically heavy NH$_4^+$ is combined with light oxygen isotopes nearing local $\delta^{18}$O-H$_2$O into the nitrate pool. This decouples the two isotopes by decreasing the $\delta^{18}$O-NO$_3$ of nitrate overall whilst increasing $\delta^{15}$N-NO$_3$.

CPND is a widespread process in Arctic shelves and has been observed in the Chukchi Sea (Brown et al., 2015; Granger et al., 2018) and the East Siberian Sea (Fripiat et al., 2018) and contributes to the observed Arctic-wide nitrogen deficit in relation to phosphate (Torres-Valdés et al., 2013; Yamamoto-Kawai et al., 2006). This shelf-derived signal is exported into the Arctic halocline (Granger et al., 2018), namely through the TPD, and can be traced in the outflowing water-masses in Fram Strait (Tuerena et al., 2021a), reflecting the impact of shelf processes on PSW nutrient ratios. Thus, the low N*, light $\delta^{18}$O-NO$_3$ and heavy $\Delta$(15-18) signal exported in the PSW is the signature of N loss on Eurasian shelves and the Chukchi Sea.

DSi concentrations in outflowing PSW are 1.2μM higher and $\delta^{30}$Si(OH)$_4$ is isotopically heavier by +0.11‰ relative to inflowing AW (Table 2). Documented Pacific and meteoric sources of DSi are isotopically light (Hawkings et al., 2017; Pokrovsky et al., 2013; Reynolds et al., 2006; Sun et al., 2018) but DSi behaves non-conservatively across the Arctic Ocean. DSi uptake by phytoplankton in the Arctic ocean and loss due to biogenic Si burial fractionate the upper water column $\delta^{30}$Si(OH)$_4$ towards heavier signatures (Brzezinski et al., 2021; Liguori et al., 2020; Varela et al., 2016).

Varela et al. (2016) suggest the heavy signal observed in the deep Arctic is sourced from intermediate Atlantic-originating waters but we observe no significant enrichment of $\delta^{30}$Si(OH)$_4$ in the intermediate water masses of Fram Strait (Figure 3, Figure 4). Given that the inflowing AWs are already too poor in DSi to contribute to isotopic enrichment, the observed increase in DSi concentrations may point to riverine DSi sources subject to enrichment due to biogenic si production and burial instead (Brzezinski et al., 2021) .

As seawater is undersaturated with respect to biogenic Si at all depths in the ocean (Archer et al., 1993), biogenic Si dissolution occurs in the water column and at the sediment-water interface. Upon burial, biogenic Si will continue to dissolve until pore-waters are saturated (Kamatani, 1982; Nelson et al., 1995). Arctic shelves are characterized by a shallow water column with

relatively high sedimentation rates influenced by river and biogenic fluxes, conditions favourable to reduced biogenic Si

exposure to dissolution, and rapid burial. Therefore, it is expected that Arctic shelf seas are particularly efficient at removing biogenic Si through opal burial. Although some studies have reported low opal burial rates and rapid recycling within the seafloor of the Barents Sea shelf (Ward et al., 2022a, 2022b), this may not be the case with shallower Eurasian shelves with higher sedimentation rates and stronger riverine influence (Kara Sea, Laptev Sea & East Siberian Sea). This suggests geochemical cycling of Si can strongly vary from one Eurasian shelf to another (Macdonald et al., 2010). In areas of low nitrate

concentrations such as the Eurasian sector of the Arctic Ocean, DSi is only partially utilised in the surface as productivity is limited by N deficit, leading to fractionation. The isotopically lighter biogenic Si is preferentially buried leaving water column $\delta^{30}Si(OH)_4$ heavy overall. This contrasts with deep Arctic basins with low productivity and long water residence times which provide opportunities for more remineralisation in the water column and at the water-sediment interface, leading to relatively small modification in the water-column $\delta^{30}Si(OH)_4$ (Brzezinski et al., 2021; Liguori et al., 2020). The heavy $\delta^{30}Si(OH)_4$

signatures of PSW thus records the partial utilisation and the loss of lighter Si through burial in the Arctic shelves.

Figure 7c shows the relationship between $\Delta(15-18)$ and $\delta^{30}Si(OH)_4$ in samples below the MLD which should not be affected by seasonal biological fractionation. A gradient is observed between AW and PSW, with a gradual increase in $\Delta(15-18)$ from 2‰ to 4‰ as salinity decreases, and an increase in $\delta^{30}Si(OH)_4$ from +1.7‰ to +2‰, linking the processes of denitrification (Fripiat et al., 2018; Granger et al., 2011, 2018) and removal of isotopically light DSi sources through biogenic Si burial in the

350 shelves (Brzezinski et al., 2021; Liguori et al., 2020) contributing to the evolution of the dual isotope signal of PSW. The combination of both CPND and biogenic Si burial indicated by the isotopic signatures of N and DSi can only occur in areas which receive a direct high influx of terrestrial DSi and hosts CPND, namely, the Bering Sea and Eurasian shelves.

AW entering the Arctic is poor in DSi which limits biological uptake in AW (Agustí et al., 2018; Krause et al., 2018, 2019). Any excess DSi (e.g from Pacific and shelf waters supplied to AW) will be consumed during the growth season until nitrate is

355 exhausted. The enrichment of $\delta^{30}Si(OH)_4$ in Arctic waters exported out of Fram Strait points towards partial utilisation of DSi, constrained by the availability of nitrate within the TPD (Brzezinski et al., 2021; Liguori et al., 2021). The combination of supply and use of these nutrients is reflected in Panel D of Figure 7, where PSW is distinct from AW with positive Si* (DSi sources from terrestrial runoff and Pacific influence where nitrate is in deficit relative to phytoplankton requirements) and heavy $\delta^{30}Si(OH)_4$ signatures whereas the relationship with salinity reflects the mixing of these distinct water mass signatures.

While AW signals remain clustered, large variability in PSW Si* and isotopic signature highlight the regional variation and complexity of the Si budget around the Arctic ocean (Table 2).

In summary, low availability of nitrogen in the Eurasian sector of the Arctic Ocean appears to regulate the extent to which DSi is utilised and subsequently exported through PSW. At Fram Strait, PSW carries the isotopic signals of DSi and N modification within Eurasian shelves through processes such as benthic denitrification and partial utilisation of DSi.

## 4.2 Si cycling in the Arctic ocean and sources of dissolved silicon exported through Fram Strait

The Arctic exports significant amounts of DSi but not N through Fram Strait (Torres-Valdés et al., 2013). Modification of PSW as a result of shelf processes can be traced across the Arctic simultaneously using N & DSi isotopes (Figure 7c). Here we use $\delta^{30}Si(OH)_4$ against ln[DSi] plots to examine the pathway of this transformation from DSi sources to Fram strait (Figure 8). Broad negative trendlines are observed in Figure 8, but on separate trendlines for AW and the pan-Arctic, with PSW in-between. Decreasing DSi and heavier $\delta^{30}Si(OH)_4$ suggest that mixed riverine and Pacific sources of DSi are transported across the Arctic towards Fram Strait through PSW.

### 4.2.1 The Bering Strait inflow

$\delta^{30}Si(OH)_4$ in the upper 100m of the water column in the North Pacific is relatively light (~ +1.5‰), with high DSi concentrations (~40μM) (Reynolds et al., 2006). Pacific-originating nutrients are strongly modified in the Bering Strait by riverine input with high DSi:N ratio from the Yukon river, by benthic denitrification, and significant biological consumption over the broad shallow shelves in the Bering & Chuchki seas. The combined processes lead to increasing $\delta^{30}Si(OH)_4$ following biological uptake and fractionation (Brzezinski et al., 2021; Giesbrecht et al., 2022). Thus the Pacific endmember measured in the Bering Strait is heavily modified by biogeochemical cycling on shelves, and Arctic inflow here has lower DSi concentrations relative to the Pacific Ocean. At lower ln(DSi) on the Pan-Arctic trendline, riverine and Pacific sources become indistinguinshable in surface water masses of the Arctic Ocean and the TPD, reflecting that mixing and biogeochemical cycling in the high Arctic homogenises both nutrient sources prior to export in PSW.

### 4.2.2 The Eurasian shelf signal

Siberian rivers have isotopically light $\delta^{30}Si(OH)_4$ from clay mineral weathering (Mavromatis et al., 2016; Pokrovsky et al., 2013; Sun et al., 2018). However, terrestrial DSi input and biological consumption occurs simultaneously on shallow Eurasian shelves. Riverine inputs support one third of the net primary productivity of the Arctic ocean (Terhaar et al., 2021), most of which occurs on the Eurasian shelves (MacDonald et al,. 2010). Phytoplankton uptake further reduces DSi concentrations and leads to isotopically heavier $\delta^{30}Si(OH)_4$. This inference follows Brzezinski et al. (2021), as it is also reflected in the TPD. Thus, in Figure 8, the broad negative trendline from riverine and Pacific sources across the TPD to Fram Strait reflects the progressive depletion of DSi through biological uptake and biogenic Si burial resulting in isotopic enrichment as it travels through the Arctic. Partial DSi utilisation modifies both the Si budget and its isotopic composition. DSi transported from Eurasian shelves through the TPD towards Fram Strait is reflected in isotopically heavy $\delta^{30}Si(OH)_4$ in PSW which aligns with the broad Rayleigh field in Figure 8.

In Fram Strait, $\delta^{30}Si(OH)_4$ fractionation involves separate trendlines for AW and PSW. The trend for AW is statistically significant at Fram strait ($R^2>0.7$) but shifted downwards indicating a distinct Atlantic source. In contrast, the PSW trendline is shifted upwards from AW towards heavier isotopic values and higher DSi concentrations, following more closely the broader

Arctic trends. In addition, larger variability in Si isotope signatures of PSW ($R^2>0.3$) reflects the combined effects of local biological uptake and mixing between Arctic and Atlantic source signatures around Fram Strait.

### 4.2.3 Glacial influence on $\delta^{30}Si(OH)_4$ exported from the Arctic Ocean via Fram Strait

Glacial and sea ice inputs have been suggested to significantly impact Arctic Si budgets (Fripiat et al., 2014; Hawkings et al., 2017), this is evaluated further in Figure 8. Isotopic studies in Greenland and Svalbard glaciers have shown isotopically light signatures with low DSi concentrations (Hatton et al., 2019). Benthic studies in SW Greenland fjords found a significant diffusive flux of isotopically light Si into overlying shelf waters (Ng et al., 2020) although export from fjords remains to be characterised. Si inputs from Greenland and Svalbard have been suggested as significant contributors to the Arctic Si budget which is exported through PSW to the North Atlantic (Hatton et al., 2019; Hawkings et al., 2017, 2018), though the glacial freshwater content of PSW at 79°N is relatively small (<13%, Stedmon et al., 2015).

In Figure 8, we show that light $\delta^{30}Si(OH)_4$ signatures from Greenland and Svalbard glacial sources also have low DSi concentrations and do not align in the Rayleigh field with the Arctic trend observed. This suggests Greenland and Svalbard glaciers are not significantly impacting the Si budget of outflowing PSW at Fram Strait. This implies in-situ studies of glacial streams in Greenland may overestimate glacial contribution of Si to Eurasian Arctic nutrient budgets. A possible explanation for this is amorphous phases of Si represent >95% of the total Si flux (Hawkings et al., 2017) and a large fraction of this may be buried in the sediments of Arctic Fjords prior to dissolution, reducing the impact of glacially-sourced DSi.

### 4.2.4 $\delta^{30}Si(OH)_4$ of sea ice

The particularly low apparent isotopic effect of the Arctic Ocean has been attributed to the influence of sea ice and sea ice diatoms drawing down DSi (Giesbrecht et al., 2022; Varela et al., 2016). Sea ice brine is heavier or equal to surrounding waters $\delta^{30}Si(OH)_4$ (Fripiat et al., 2007, 2014) and may contribute to the isotopically heavy signature of polar waters (Liguori et al., 2020; Varela et al., 2016). A recent study from Brzezinski et al. (2021) did not find direct evidence of such an impact on a basin-wide scale. Here we evaluate the role of sea-ice in influencing Arctic $\delta^{30}Si(OH)_4$ signatures in a region influenced by brine rejection. In Figure 9, we present hydrography and late-summer profiles of DSi and $\delta^{30}Si(OH)_4$ collected from Ile-de-France section between 2017-2019 (section location is shown in Figure 1). This area is characterized by perennial sea ice cover (Schneider and Budeus, 1995). A PSW layer extends down to 125m of the water column and is influenced by brine released during winter sea ice formation (Budeus and Schneider, 1995). In the freshwater layer, a small peak in DSi concentration (2-3 μM) is observed at ~40m. The small increase in DSi concentration at this depth suggests a DSi source from sea ice processes. However, there is no distinct isotopic enrichment (Figure 9) associated with this source. Thus DSi inputs from sea ice cannot be the reason for enriched $\delta^{30}Si(OH)_4$ signatures of PSW. This inference is consistent with studies suggesting that sea ice and sea ice brine tend to be relatively low in DSi (Fripiat et al., 2017), with no significant impact on pan-Arctic isotope signatures (Brzezinski et al., 2021).

### 4.2.5 Processes affecting the export of Arctic DSi to the Atlantic Ocean

Figure 8 reveals that DSi exported to the Atlantic in PSW in 2018 was sourced from incomplete utilisation of DSi on Eurasian shelves. This leads to the question as to what limits the complete utilisation of DSi in the Arctic. In Figure 10, we plot $\delta^{15}$N-NO$_3$ versus DSi:N ratios, excluding measurements within the MLD at Fram Strait to remove seasonal uptake trends (this was not applied to measurements in the TPD and shelf seas due to the shallow nature of the water masses). The figure reveals the three mixing components of the Arctic N budget, namely, the very heavy $\delta^{15}$N-NO$_3$ values generated on the the shelves with high DSi:N ratios due to removal of light nitrate by CPND; The input of terrestrial riverine N with relatively light $\delta^{15}$N-NO$_3$ signatures ($\delta^{15}$N-NO$_3$ = 2.3‰, Francis, 2019) with variable but high DSi:N ratios; and Atlantic sources ($\delta^{15}$N-NO$_3$ = 4.8 ± 0.2‰, Tuerena et al., 2015, and DSi:N = 0.6, World Ocean Database, 2013). AW sources become important in Fram strait and contribute to nitrate by mixing across the halocline in basins where AAW underlies below PSW. Pan-Arctic N isotopic trends, shown in Figure 10, are dominated by mixing of sources rather than fractionation by biological uptake; a striking constrast to the $\delta^{30}$Si(OH)$_4$ trend (Figure 8). This is arguably caused by the near-complete utlisation of nitrate on Eurasian shelves and above the halocline, leading to limited overall fractionation from source signatures. This widespread nitrate limitation in the Arctic is attributed to fixed N loss from benthic denitrification on the shallow shelves which constitute approximately 50% of the Arctic ocean area. A significant portion of N loss from denitrification is derived from organic matter from the overlying water column (Mctigue et al., 2016; Tuerena et al., 2021c); leading to a net deficit of N exported out of the Arctic (Torres-Valdés et al., 2013; Yamamoto-Kawai et al., 2006).

Furthermore, Arctic rivers are a larger source of DSi than N (Holmes et al., 2012) and the N supplied is quickly removed in river deltas (Tuerena et al., 2021c). For example, on the Laptev sea shelves, it is estimated that 62-76% of riverine dissolved organic nitrogen is removed within a couple of months by denitrification and biolgical utilisation (Thibodeau et al., 2017). This is evident from the very low nitrate concentrations in the TPD and high DSi:N ratios (~5μM and 1.8 respectively, Doncila, 2020) which is heavily influenced by riverine inputs and modification over the Eurasian shelves. The near-absence of nitrate in surface waters overall contributes to the higher DSi:N output observed in PSW in Fram Strait. We conclude that incomplete utilisation of DSi in the Arctic ocean and its subsequent export through the Fram Strait is governed largely by widespread N limitation due to the rapid removal of nitrate in the Arctic ocean, namely on Eurasian shelves.

### 4.2.6 Evaluating contribution of DSi sources at Fram Strait

With decreasing ln(DSi) in Figure 8, riverine and Pacific sources of $\delta^{30}$Si(OH)$_4$ form a homogenous pan-Arctic trendline driven by partial utilisation of DSi, separate from AW in the Rayleigh field. The $\delta^{30}$Si(OH)$_4$ of PSW plots between these two trendlines, as a mixture of AW and Arctic-sourced nutrients instead (figure 6 & figure 8).

As the pan-Arctic relationship is strong ($R^2$=0.67), the extent of utilisation of DSi sources and their relative contribution to PSW can be estimated from the apparent pan-Arctic isotopic effect $^{30}\varepsilon$ (regression of trendline in the Rayleigh field). The two models are displayed on Figure 11. Considering the multiple nutrient pathways and physical mixing as waters are transported

from Arctic shelves to the Fram Strait in the Arctic Ocean, it is expected that the pan-Arctic dataset fits an open system. This is what is observed on Figure 11. The open model is more coherent with a $^{30}\varepsilon$ close to global estimates of -1‰ and a stronger $R^2$ for the open system model ($R^2 = 0.83$) than for a closed system ($R^2 = 0.67$). Using the pan-Arctic dataset we estimate $^{30}\varepsilon =$ -0.58‰ for closed system fractionation and $^{30}\varepsilon = $ -1.09‰ for open system fractionation (Figure 11, supplementary material S.4). This is in close agreement with measured isotopic effects in the Canadian Arctic sector (-0.59‰ and -1.19‰ for closed and open systems respectively, Giesbrecht et al., 2022) with may be lower than the global average due to sea-ice diatoms, but also dilutive effects.

Using the PSW signature calculated in Table 2 (1.84 ± 0.09‰), the apparent remaining nutrient fraction ($f_{PSW}$) within PSW can be estimated from the isotopic effects calculated above. We estimate $f_{PSW} = 0.37 \pm 0.06$ and $f_{PSW} = 0.51 \pm 0.08$ for closed and open systems respectively. Considering the large-scale patterns of transport, circulation and mixing within the Arctic Ocean, we can assume the system is open as nutrients are likely to be frequently resupplied and closed system assumptions would lead to an overestimation of nutrient consumption, but highlight that care needs to be taken when applying fractionation models to the open ocean. Modelled DSi within PSW in an open system would thus be 7.8 ± 0.6 µM and 1.89 ± 0.02‰ (Based on parameters detailed in supplementary material S.4).

Based on the calculation above, riverine sources contribute to 40 ± 4% of the total DSi inventory at Fram Strait, with Pacific sources contributing to 8 ± 1%. Although sea ice has been proved to play an important role for DSi cycling in other parts of the Arctic Ocean (Giesbrecht et al., 2022; Liguori et al., 2021) this calculation assumes sea ice only dilutes ambient DSi concentrations and has no net isotopic effect based on our observation locally (section 4.2.4). Nevertheless, this basic calculation produces a rough estimate based on DSi concentrations and isotopic signatures measured within PSW. It illustrates two things (1) that a mixture of heavily utilised riverine DSi and partially utilised Pacific-originating nutrients control the nutrient inventory exported through the PSW; (2) PSW DSi inventory is highly sensitive to the extent of utilisation of riverine DSi on Arctic shelves due to the high initial concentration. This calculation highlights that modification of riverine nutrient sources and removal on Arctic shelves is likely to have a large influence on the Atlantic DSi export and its isotopic budgets. To improve the above estimate, accurate understanding of DSi consumption on shelves prior to export in the central Arctic Ocean is required, with improved isotopic signature determination of riverine sources and shelf water masses.

**5 Future implications**

Our results highlight some important connections between nutrient cycling and the control on the exchange of nutrients between the Arctic and the Atlantic Ocean. This study has identified a link between the Arctic N and Si cycles: low Nitrogen availability regulates the extent of DSi drawdown in exported PSW and is traced to Arctic shelf processes. The nitrogen deficit is generated by biological Arctic processes such as CPND. This along with the extent of utilisation of DSi sources controls the excess DSi exported out of the Arctic ocean through gateways such as the Fram Strait. In the changing Arctic ocean, this has far-reaching implications to ecosystems and nutrient budgets as discussed below.

Using $\delta^{30}Si(OH)_4$ signatures, we have estimated that over 40% of DSi exported out through Fram Strait is of riverine origin. Freshwater inputs to the Arctic ocean from the Eurasian sector are expected to increase in response to climate change (Mcclelland et al., 2006; Rawlins et al., 2010). Increasing riverine discharge and permafrost degradation is accelerating the transport of terrestrial material to Eurasian shelves and likely increasing the export of major nutrients (Zhang et al., 2021). As

$NO_3$ delivery from rivers is low, riverine sources of DSi are increasing faster than N inputs.

Light, DSi and $NO_3$ availability all play an important role in dictating the complex patterns of diatom production around the Arctic Ocean (Giesbrecht et al., 2019; Krause et al., 2019). Our study illustrates that $NO_3$ availability plays an important role for biogenic Si production in the Eurasian Arctic. Nitrogen is quickly removed in Siberian rivers at low salinities (Sanders et al., 2021; Tuerena et al., 2021c) through benthic denitrification, with roughly 70% of terrestrial N removed before reaching

the seawater endmember (Letscher et al., 2013) depleting N in relation to DSi in the deeper water column. Such rapid N removal implies additional terrestrial $NO_3^-$ inputs are not likely to significantly impact N-availability. Nitrogen deficiency on Arctic shelves is currently limiting DSi consumption to only 14.3% of its net riverine input (Le Fouest et al., 2013). This implies that as terrestrial DSi inputs increase, a larger proportion of terrestrial DSi will remain unutilised and ultimately get transported through the TPD out to Fram Strait. This will increase the export of DSi to the North Atlantic, but also alter the

$\delta^{30}Si(OH)_4$ of PSW which is derived from the partial biological utilisation of DSi. Terrestrial DSi inputs increasing in the future combined with increased N-limitation will reduce the percentage of DSi consumption in the Arctic ocean, leading to lighter isotopic signatures of DSi exported towards the North Atlantic.

Locally, the larger export of DSi through the TPD has implications for nutrient dynamics in Fram Strait. N-limitation is strong in PSW and is predicted to increase in AW (Tuerena et al., 2021a). Increasing primary production in the Arctic shelves as sea

ice melts and light availability increases (Arrigo et al., 2008; Arrigo and van Dijken, 2015) will increase N-demand, and further N losses through denitrification which could reduce DSi uptake further and limit net productivity from silicifying species and impact carbon drawdown in Fram Strait. A decline in diatoms and a shift towards smaller phytoplankton assemblages is already observed with warming in Fram Strait (Lalande et al., 2013). Such changes will be accentuated further with N-limitation.

We also recognize there are competing influences on the future nutrient status of the Fram Strait. The higher export of DSi can

compensate for decreasing DSi supply through AW to the Arctic ocean resulting from Atlantification (Arthun et al., 2012; Lind et al., 2018) which leads to decreasing DSi concentrations in AW (Hátún et al., 2017). While terrestrial increase in DSi input and reduced utilisation in the Arctic will supersede this signal in PSW over time, this can potentially lead to a decrease in the DSi inventory of intermediate and deep waters of the Arctic ocean influenced by AW, while increasing DSi export out of the Arctic through the PSW.

The far-reaching consequences of the predicted future increases in Arctic DSi export to the North Atlantic imply changes to primary production patterns and DSi concentrations in deep water masses formed here. Waters in the North Atlantic are richer in nitrate than DSi, and available evidence indicate DSi limitation of diatom spring blooms due to limiting concentrations of silicic acid in the region (Henson et al., 2006; Leblanc et al., 2005). This envisioned additional supply of DSi can impact the duration of diatom blooms in the sub-Arctic North Atlantic (Allen et al., 2005), and possibly enhance diatom production with

subsequent implications for carbon export to the deep ocean. In longer time scales, this can also increase the preformed DSi inventories in the North Atlantic deep waters, with an impact on the nutrient status of the deep water masses worldwide.

## 6 Conclusions

Previous understanding of the importance of physical (water-mass mixing) vs biological (production and dissolution) controls in setting $\delta^{30}Si(OH)_4$ distribution across the Arctic was limited by the lack of direct measurements at major gateways
(Brzezinski et al., 2021). This study provides the first full depth profiles of $\delta^{30}Si(OH)_4$ in Fram Strait, in combination with $\delta^{15}N$-$NO_3$ and $\delta^{18}O$-$NO_3$, closing gaps in the Arctic isoscape and confirming mechanisms of transformation.

Isotopic measurements document the transformation of PSW outflowing through Fram Strait, with isotopic signatures $\Delta(15\text{-}18) = 4.22 \pm 0.89‰$ and $\delta^{30}Si(OH)_4 = +1.85 \pm 0.09‰$. $\delta^{30}Si(OH)_4$ is significantly enriched by $+0.11‰$ in PSW compared to inflowing AW, while $\Delta(15\text{-}18)$ is enriched by $1.37‰$, showing significant source modification of the nutrients between the
inflow and outflow waters.

Further examination of DSi & N isotopes trace nutrient sources and modification processes in PSW primarily to Eurasian shelves: The increase in DSi concentration and enrichment of $\delta^{30}Si(OH)_4$ is traced to biological uptake of DSi and partial burial of biogenic Si on the shelves, sustained by the high DSi load from Eurasian rivers. Export of DSi out of the Arctic through Fram Strait is ultimately regulated by N-limitation resulting from N-poor input from terrestrial sources combined with efficient
removal of N through assimilation and denitrification on shelves. This is documented in PSW through de-coupling of the oxygen and nitrogen isotopes of nitrate from traditional 1:1 relationship. Glacial influence from Greenland and Svalbard glaciers and Pacific inflow appeared of smaller influence at Fram Strait in PSW, with riverine sources contributing to ~40% of the DSi exported out of Fram Strait.

The measurement of DSi & N isotopes provides the first insights into the coupling of the N and Si cycle in the Arctic Ocean.
Nitrate limitation during primary production generates excess DSi which is subsequently exported to the North Atlantic. As riverine nutrient sources of DSi are expected to increase faster than N with climate warming, this can enhance N limitation within the Eurasian Arctic ocean and increase the export of DSi to the North Atlantic ocean.

**Funding.** This work was supported by a Natural Environment Research Council (NERC) Doctoral Training Partnership grant
(NE/L002558/1) and from the ARISE project NE/P006310/1 awarded to Raja S. Ganeshram, part of the Changing Arctic ocean programme, jointly funded by the UKRI NERC and the German Federal Ministry of Education and Research (BMBF).

**Data availability.** Nutrient (doi:10.5285/b61d58df-b8e8-11c4-e053-6c86abc0246c, Brand et al., 2020) and nitrate isotope data (doi:10.5285/b93fb7c0-110e- 2470-e053-6c86abc05d60, Tuerena and Ganeshram, 2021a) for JR17005 are publicly
available from the British Oceanographic Database website. Silicon isotope data will be made publicly available on the British Oceanographic Database website (JR17005) and the Norwegian Polar Institude Data repository (FS2017-FS2919) upon acceptance of the manuscript.

**Author contribution.** MCFD wrote the manuscript. MCFD, LP, RET, PAD and RSG designed the study. MCFD, RET and AD analysed nitrate isotope samples. MCFD and LP analysed silicon isotope samples. All authors contributed to field work implementation and to the final version of the manuscript.

**Competing interests.** The authors declare that they have no conflict of interest.

**Acknowledgements.** We thank the crew and participants of Changing Arctic ocean cruises onboard the RRS James Clark Ross and Fram Strait Arctic Outflow Observatory cruises onboard RV Lance and RV Kronprins Haakon for support in sampling. We also thank the ARISE team for the collaborative sampling effort and sharing scientific ideas.

**Table 1: Summary of sections along which samples were collected**

| Year | Cruise | Dates | Vessel | Section |
|------|--------|-------|--------|---------|
| 2018 | JR17005 | 9 May - 9 June | RRS James Clark Ross | Fram Strait (79˚N) |
| 2018 | FS2018 | 25 August – 11 September | RV Kronprins Haakon | Fram Strait (78° 50'N) Ile de France |
| 2017 | FS2017 | 24 August -13 September | RV Lance | Ile de France |
| 2019 | FS2019 | 1 September – 16 September | RV Kronprins Haakon | Ile de France |

**Table 2: Averaged water mass signatures of the Fram Strait (a) excluding the mixed layer depth and (b) within the mixed layer in spring ( JR17005) and summer (FS2018). Water mass definitions based on (Richter et al., 2018). AW = Atlantic Water, PSW = Polar Surface Water, wPSW = warm PSW, AAW = Arctic Atlantic Water, DW = Deep Water, DSOW = Denmark Strait Overflow Water. N\* is defined as N\*= NOx – 16\*PO$_4^-$) and Si\* is defined as Si\* = Si(OH)$_4^-$ - NO$_x$).**

| | Nitrate (µM) | N* | Si* | δ$^{15}$N-NO3 (‰) | δ$^{18}$O-NO3 (‰) | Δ(15-18) (‰) | DSi (µM) | δ$^{30}$Si(OH)$_4$ (‰) |
|------|------|------|------|------|------|------|------|------|
| AW | $12.10 \pm_{0.98}$ | $-0.55 \pm_{0.38}$ | $-6.73 \pm_{0.95}$ | $5.18 \pm_{0.21}$ | $2.33 \pm_{0.51}$ | $2.85 \pm_{0.46}$ | $5.42 \pm_{0.71}$ | $+1.74 \pm_{0.06}$ |
| PSW | $8.08 \pm_{2.19}$ | $-2.36 \pm_{0.70}$ | $-1.37 \pm_{2.43}$ | $5.44 \pm_{0.15}$ | $1.22 \pm_{0.92}$ | $4.22 \pm_{0.89}$ | $6.64 \pm_{1.71}$ | $+1.85 \pm_{0.09}$ |
| wPSW | $11.52 \pm_{1.59}$ | $-0.50 \pm_{1.14}$ | $-5.00 \pm_{2.61}$ | $5.09 \pm_{0.34}$ | $2.12 \pm_{0.78}$ | $2.97 \pm_{0.84}$ | $6.53 \pm_{1.52}$ | |
| AAW | $11.94 \pm_{1.40}$ | $-1.19 \pm_{0.59}$ | $-5.26 \pm_{1.37}$ | $5.31 \pm_{0.33}$ | $1.93 \pm_{0.77}$ | $3.38 \pm_{0.61}$ | $6.63 \pm_{1.31}$ | $+1.74 \pm_{0.06}$ |
| DW | $14.02 \pm_{1.11}$ | $-0.98 \pm_{0.47}$ | $-4.58 \pm_{1.66}$ | $5.28 \pm_{0.17}$ | $1.60 \pm_{0.33}$ | $3.68 \pm_{0.30}$ | $9.45 \pm_{2.48}$ | $+1.65 \pm_{0.13}$ |
| DSOW | $12.33 \pm_{1.18}$ | $-0.90 \pm_{0.55}$ | $-6.20 \pm_{1.08}$ | $5.24 \pm_{0.20}$ | $1.91 \pm_{0.69}$ | $3.33 \pm_{0.65}$ | $6.15 \pm_{1.00}$ | $+1.75 \pm_{0.08}$ |
| | Nitrate (µM) | N* | Si* | δ$^{15}$N-NO3 (‰) | δ$^{18}$O-NO3 (‰) | Δ(15-18) (‰) | DSi (µM) | δ$^{30}$Si(OH)$_4$ (‰) |
| AW spring | $3.83 \pm_{4.49}$ | $-2.06 \pm_{0.76}$ | $-0.73 \pm_{3.57}$ | $8.48 \pm_{2.34}$ | $6.70 \pm_{3.29}$ | $1.78 \pm_{1.19}$ | $3.19 \pm_{1.20}$ | $+2.04 \pm_{0.37}$ |
| AW summer | $2.71 \pm_{1.24}$ | $-1.27 \pm_{0.41}$ | $-1.76 \pm_{0.31}$ | $6.63 \pm_{2.02}$ | $5.23 \pm_{3.22}$ | $1.39 \pm_{1.48}$ | $1.03 \pm_{0.98}$ | |
| PSW spring | $1.50 \pm_{1.18}$ | $-5.52 \pm_{2.70}$ | $2.73 \pm_{2.73}$ | $8.80 \pm_{1.06}$ | $4.63 \pm_{1.49}$ | $4.17 \pm_{1.62}$ | $4.28 \pm_{2.93}$ | $+2.12 \pm_{0.29}$ |
| PSW summer | $0.19 \pm_{0.12}$ | $-6.03 \pm_{2.21}$ | $1.06 \pm_{1.14}$ | | | | $1.26 \pm_{1.11}$ | |

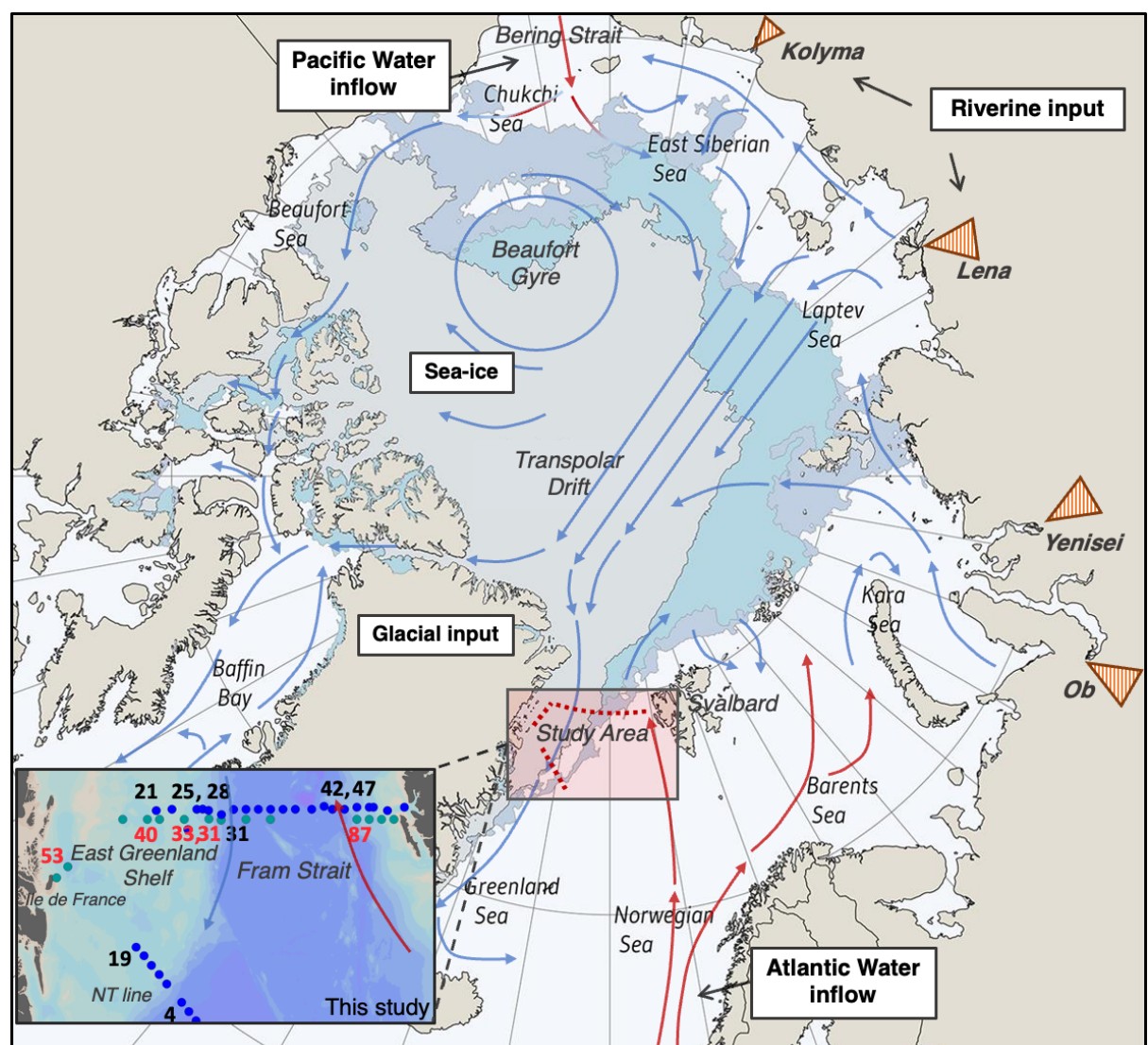

**Figure 1: Map of the Arctic ocean showing the study area of this research and general surface circulation patterns within the Arctic ocean. Red arrows represent warm, saline currents of the Atlantic and Pacific, and blue arrows represent fresh, cold water modified within the Arctic ocean (adapted from Tremblay et al., 2005). Orange triangles show the river deltas of four rivers which constitute the largest freshwater source to the Atlantic-Arctic sector: the Ob, Yenisei, Lena & Kolyma rivers. Shaded areas of the central Arctic shows September sea ice extent for 2006 (dark blue), 2017 (light blue) and 2020 (grey-blue). Figure adapted from NSIDC, 2020. Inset: Nitrate isotope sample stations for this study are shown with blue dots (JR17005) and green dots (FS2018). The station numbers for silicon isotope profiles measured within this study are shown in red for FS2018 & black for JR17005.**

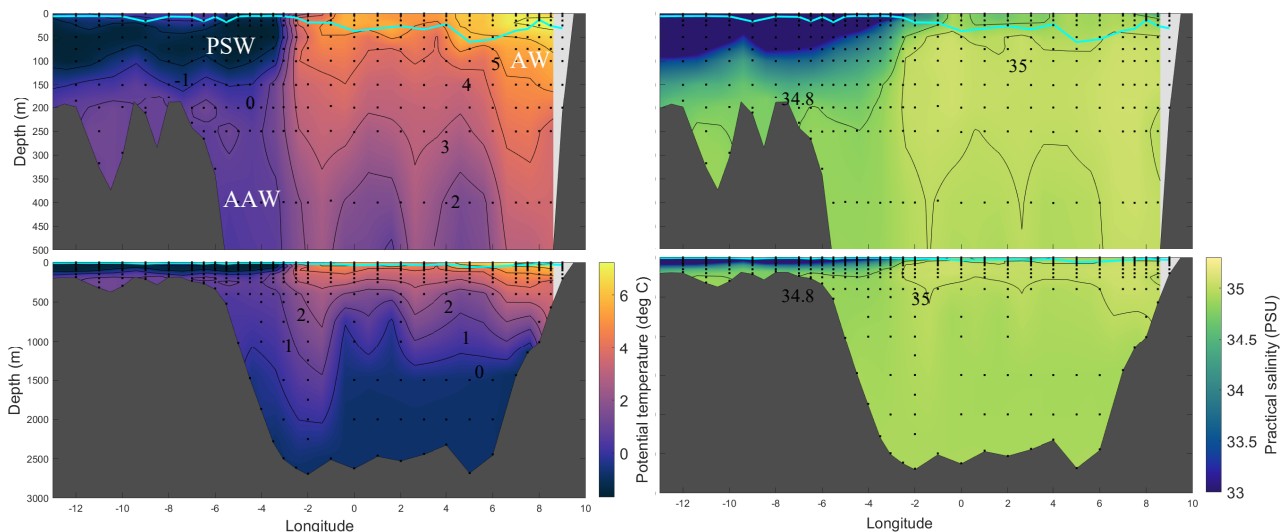

**Figure 2: Hydrography of Fram Strait cruise FS2018 from August-September 2018 presented for full depth (bottom panel) and
610 upper water column (top panel) for temperature (left) and salinity (right). The mixed-layer depth is shown by a cyan line (calculation of MLD is described in method section below). Isotherms (left) & isohalines (right) are also displayed. Atlantic Water (AW), Polar Surface Water (PSW) & Arctic Atlantic Water (AAW) are marked.**

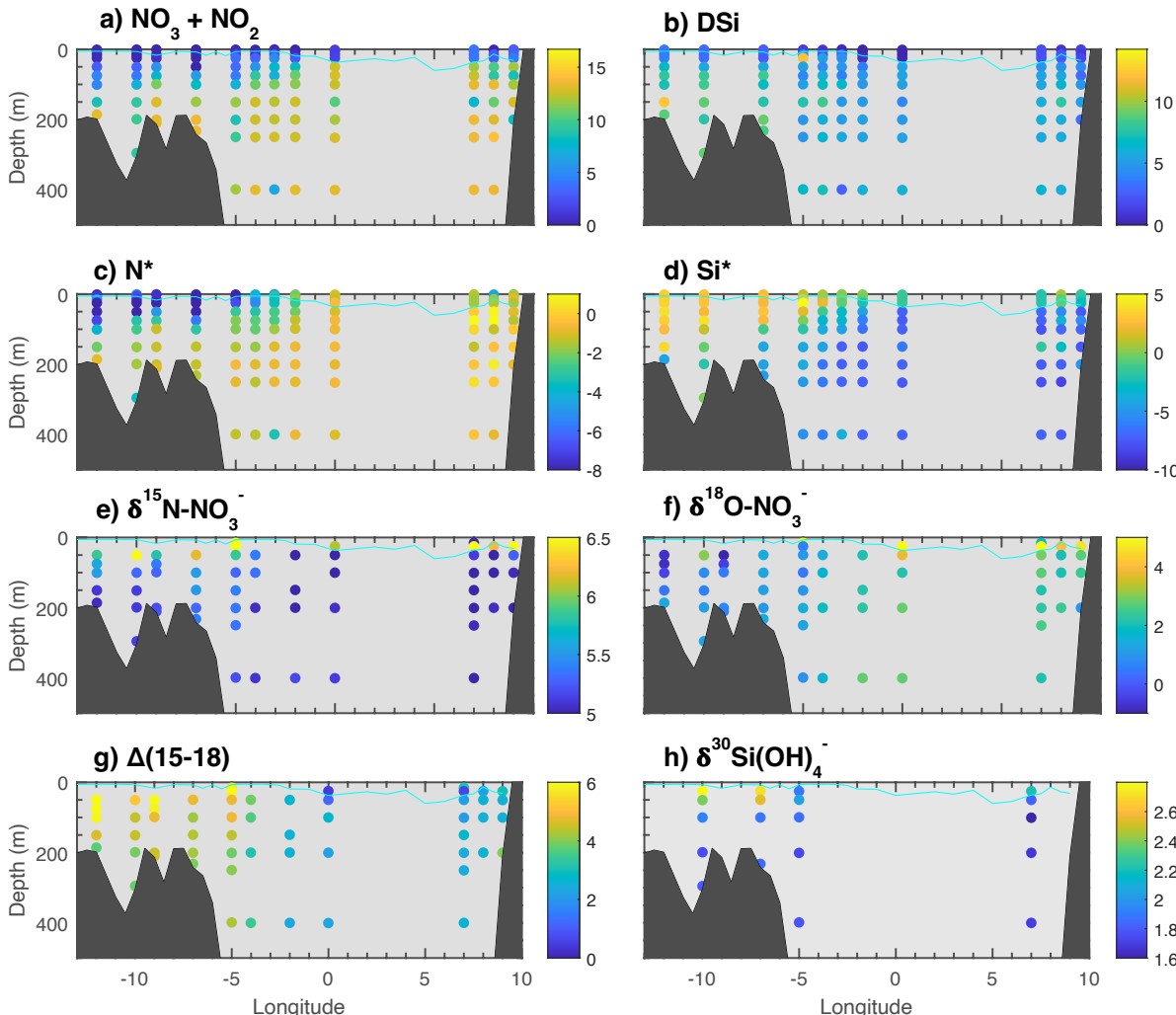

**Figure 3: Nutrients and isotopes across Fram Strait section of late summer 2018 a) NOx (where NOx= nitrate + nitrite), b) DSi, c) N\* (where N\*= NOx – 16\*PO$_4^-$), d) Si\* (where Si\* = Si(OH)$_{4-}$ - NOx), e) δ$^{15}$N-NO$_3$, f) δ$^{18}$O-NO3, g) Δ(15-18) (where Δ(15-18) = δ$_{15}$N-NO$_3$ - δ$^{18}$O-NO$_3$), h) δ$^{30}$Si(OH)$_4$. Cyan line displays MLD for the section (calculation described in method section).**

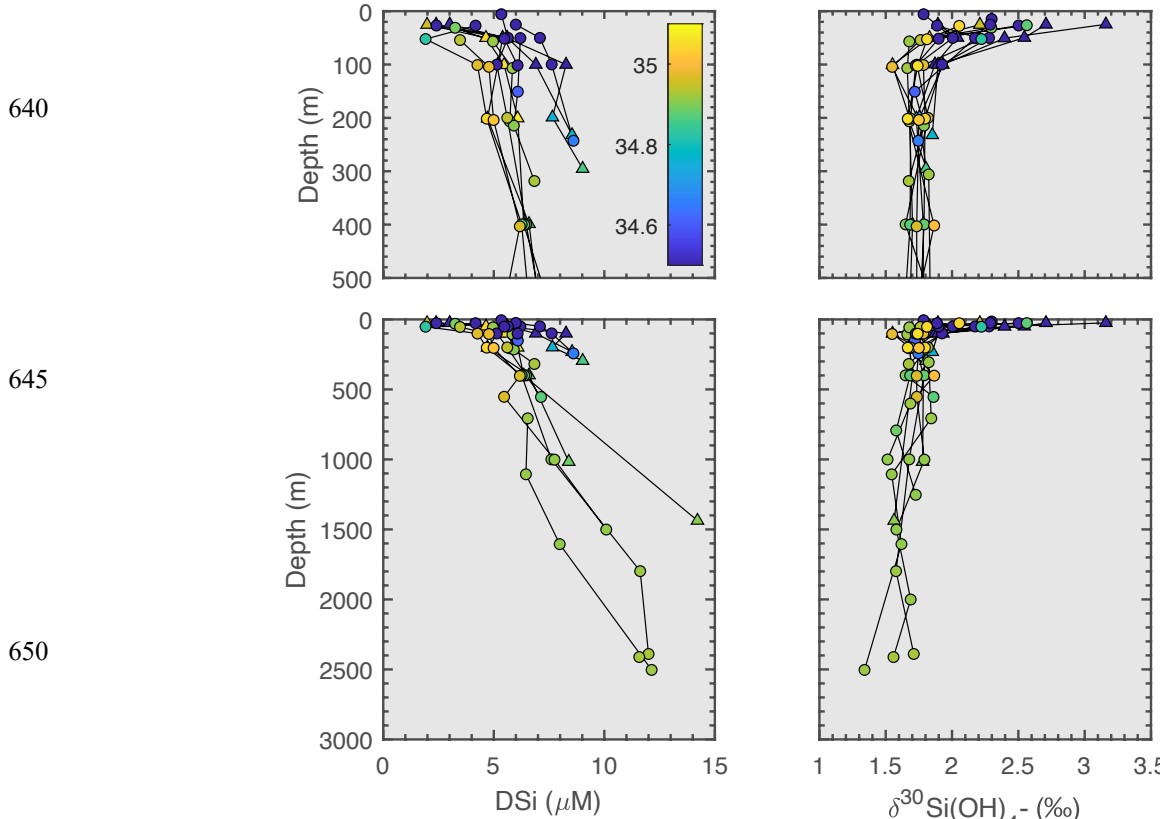

Figure 4: DSi concentrations (left) and dissolved silicon isotope profiles (right) for spring (JR17005, circles) and late summer
(FS2018, triangles) of the 2018 growth season in Fram Strait. Colour scale represents salinity (psu).

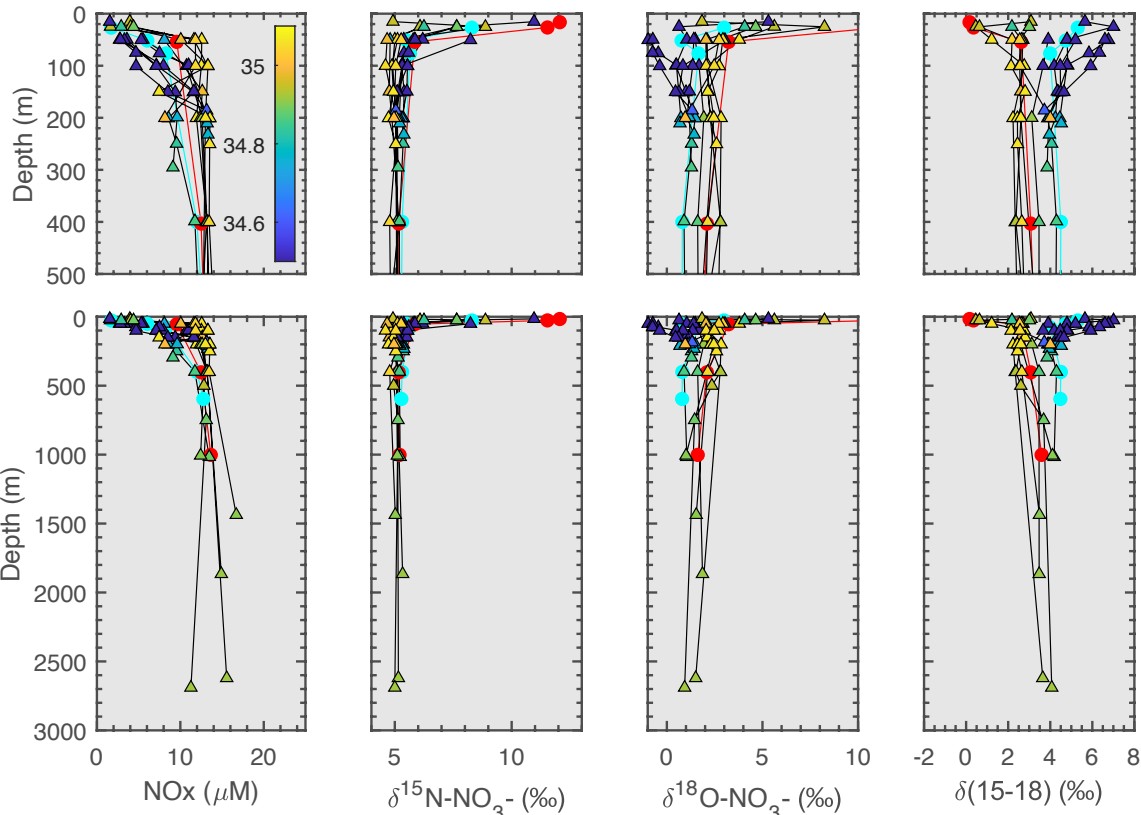

**Figure 5: Nitrate concentrations (left), δ¹⁵N-NO₃ (middle-left), δ¹⁸O-NO₃ (middle-right) and Δ(15-18) profiles (right) for FS2018 (triangles) in Fram Strait. Typical profiles for PSW (cyan) and AW (red) from JR17005 are shown in circles for comparison between studies (Tuerena et al., 2021). Colourscale represents salinity (psu).**

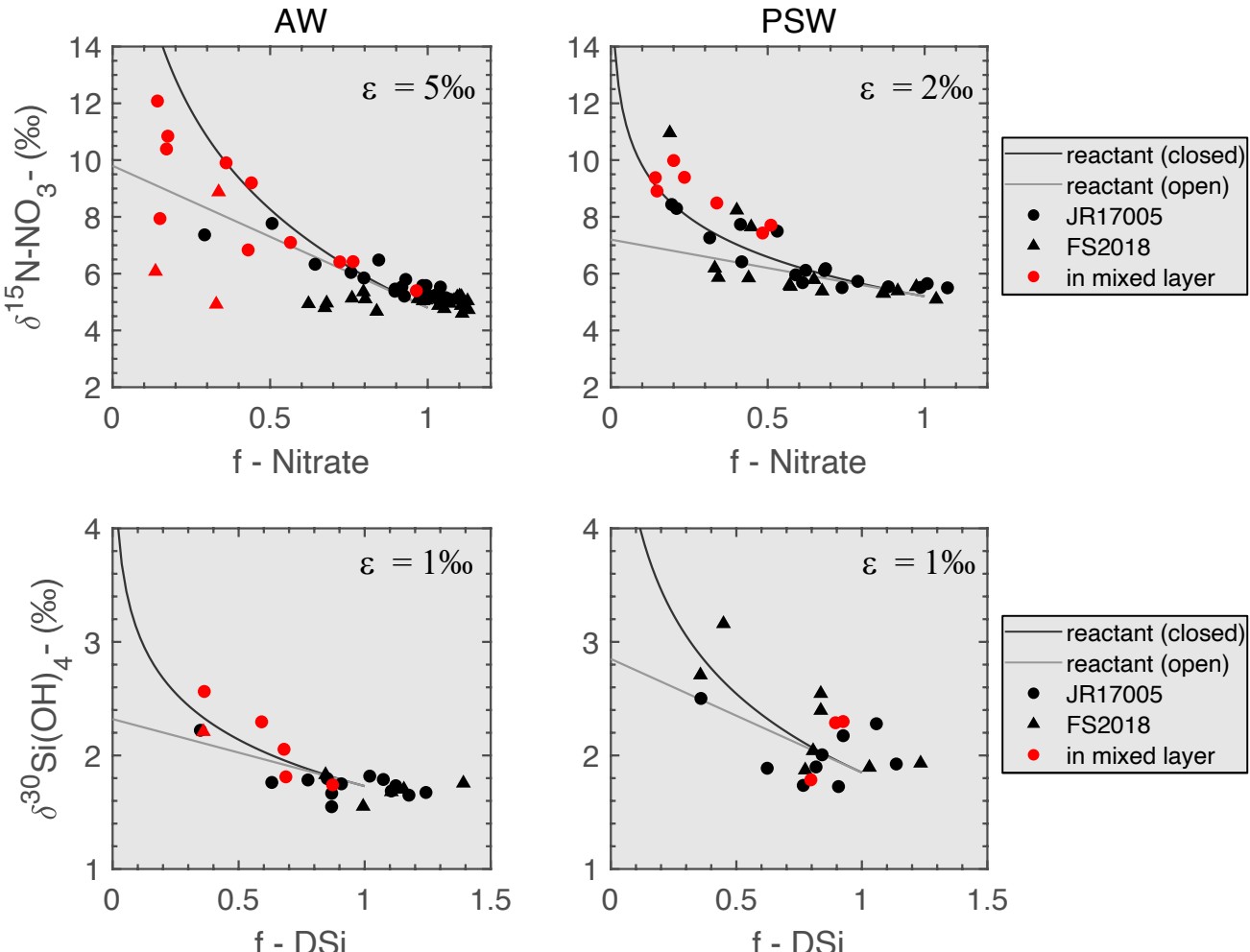

**Figure 6: Top panels: nitrate utilisation vs δ¹⁵N-NO₃ for AW (left, depth <600m) and PSW (right, depth < 150m). Bottom panels: DSi utilisation vs δ³⁰Si(OH)₄ for AW (left, depth <600m) and PSW (right, depth < 150m). Circles are denote measurements from JR17005 (spring) and triangles from FS2018 (summer). Red symbols show measurements within the mixed layer. Black line follows the closed fractionation model and grey line an open fractionation model. f is the fraction of nutrient remaining, calculated from the nutrient concentrations of water masses AW and PSW below the MLD (Table 2).**

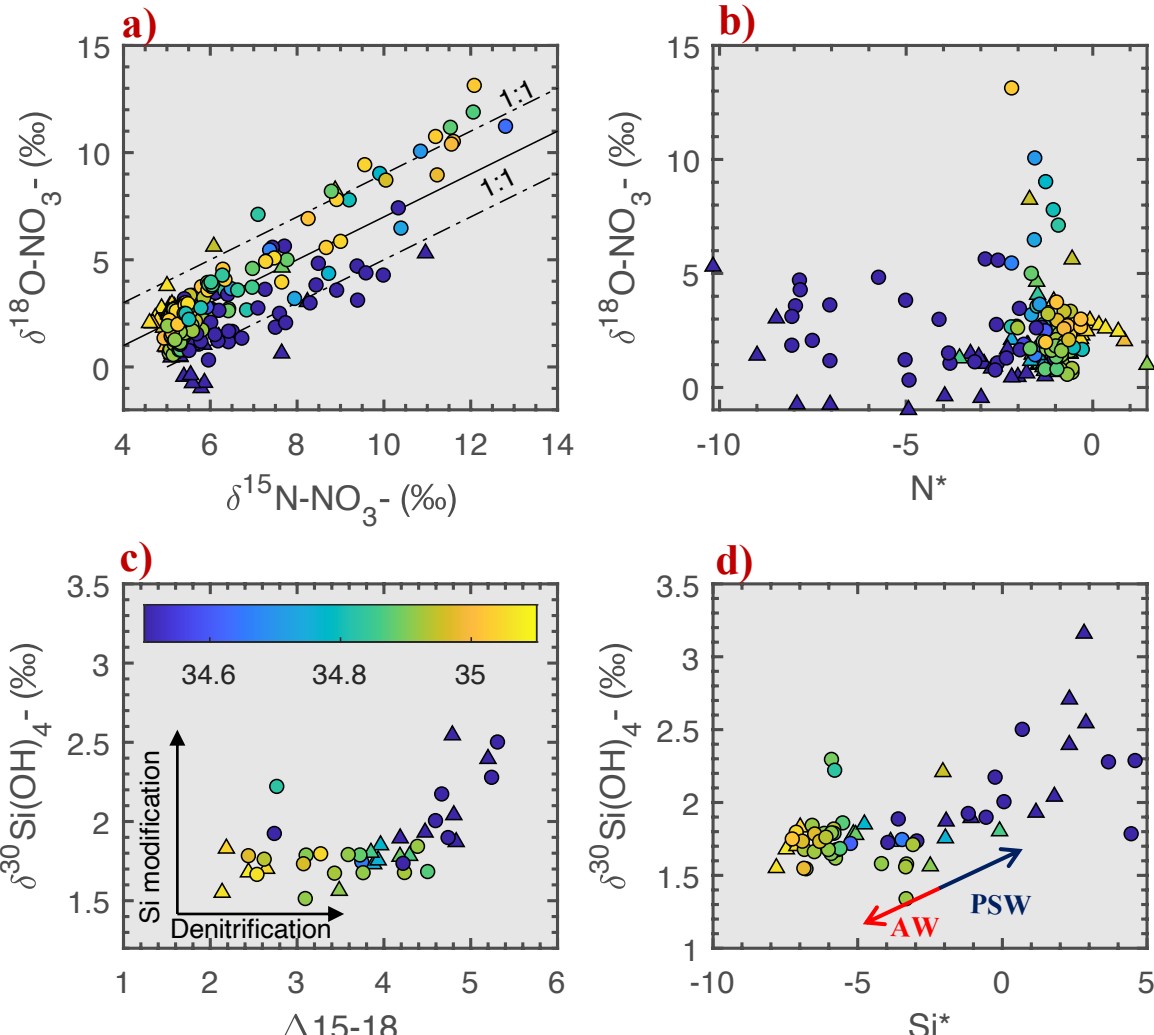

**Figure 7: Fram Strait measurements of a) δ15N-NO3 vs δ18O-NO3 (solid and dotted lines show 1:1 fractionation lines). b) δ18O-NO3 vs N*. c) Δ(15-18) vs δ30Si(OH)4 excluding samples from within the mid-layer depth to remove seasonal variation. d) Si* vs δ30Si(OH)4. In all figures, circles represent spring (JR17005) and triangles show late summer (FS2018). Colorscale for all plots show salinity (psu).**

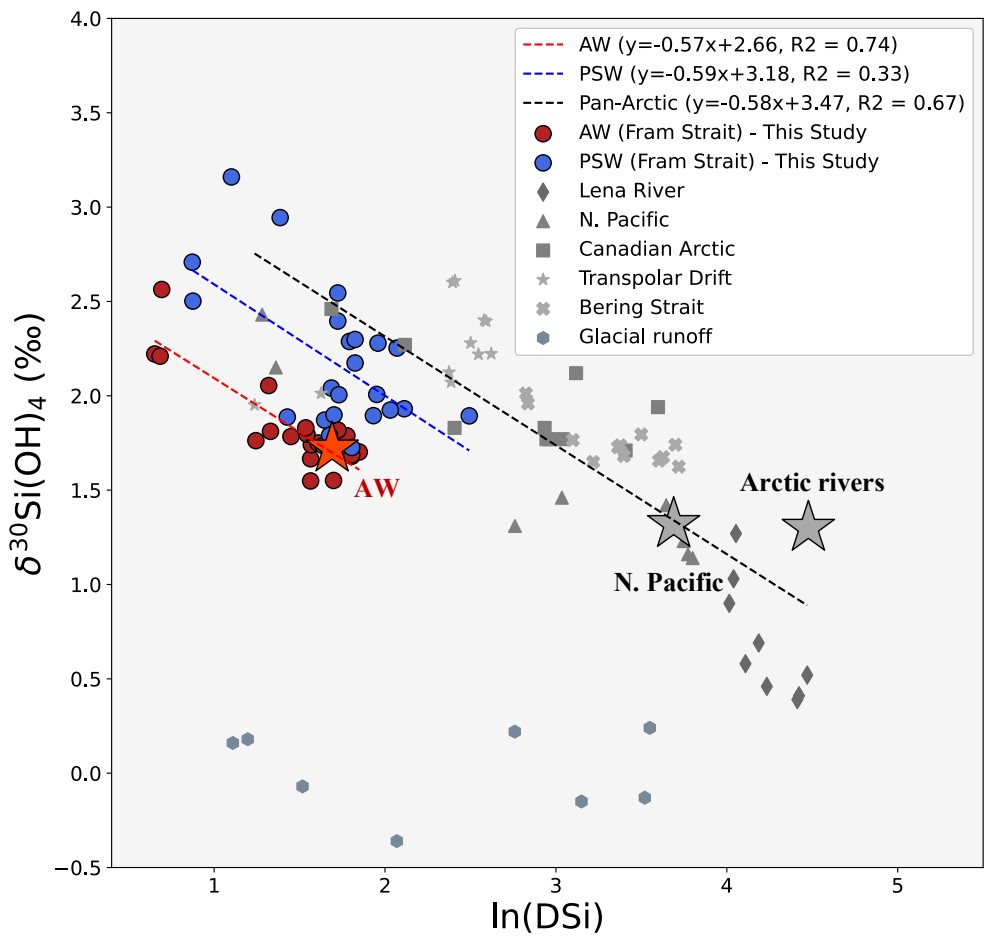

**Figure 8: Pan-Arctic trends of $\delta^{30}Si(OH)_4$ against ln(DSi).** Colored dots show measurements from within AW (red, max. depth = 600m) and PSW watermasses (blue, max.depth = 150m) from this study based on water mass definitions in Table 2. Grey symbol sets are published $\delta^{30}Si(OH)_4$ from major DSi sources to the surface Arctic domain and surface water masses. Triangles: N.Pacific (<100m, stations 1-6); Stars: Transpolar drift (<60m, stations 30-38 from Brzezinski et al., 2021). Crosses: Bering Strait (max. depth = 60m, stations 4-6 from Brzezinski et al., 2021). Squares: Canadian Arctic (surface and intermediate water mass signatures of the Canadian Arctic sector, from Table 2 in Giesbrecht et al., 2022). Octogones: Glacial runoff from Greenland and Svalbard glaciers (Hatton et al., 2019). Diamonds: Lena river (Sun et al., 2018). Stars show average endmember composition of AW (red) and Pacific and riverine sources (Grey). Red dotted trendline is the least-squared regression for $\delta^{30}Si(OH)_4$ vs the natural logarithm of DSi within AW, blue and grey dotted trendlines are the equivalent for PSW and pan-Arctic (excluding Fram Strait) respectively. These trendlines show fractionation from partial utilisation of DSi consistent with fractionation models.

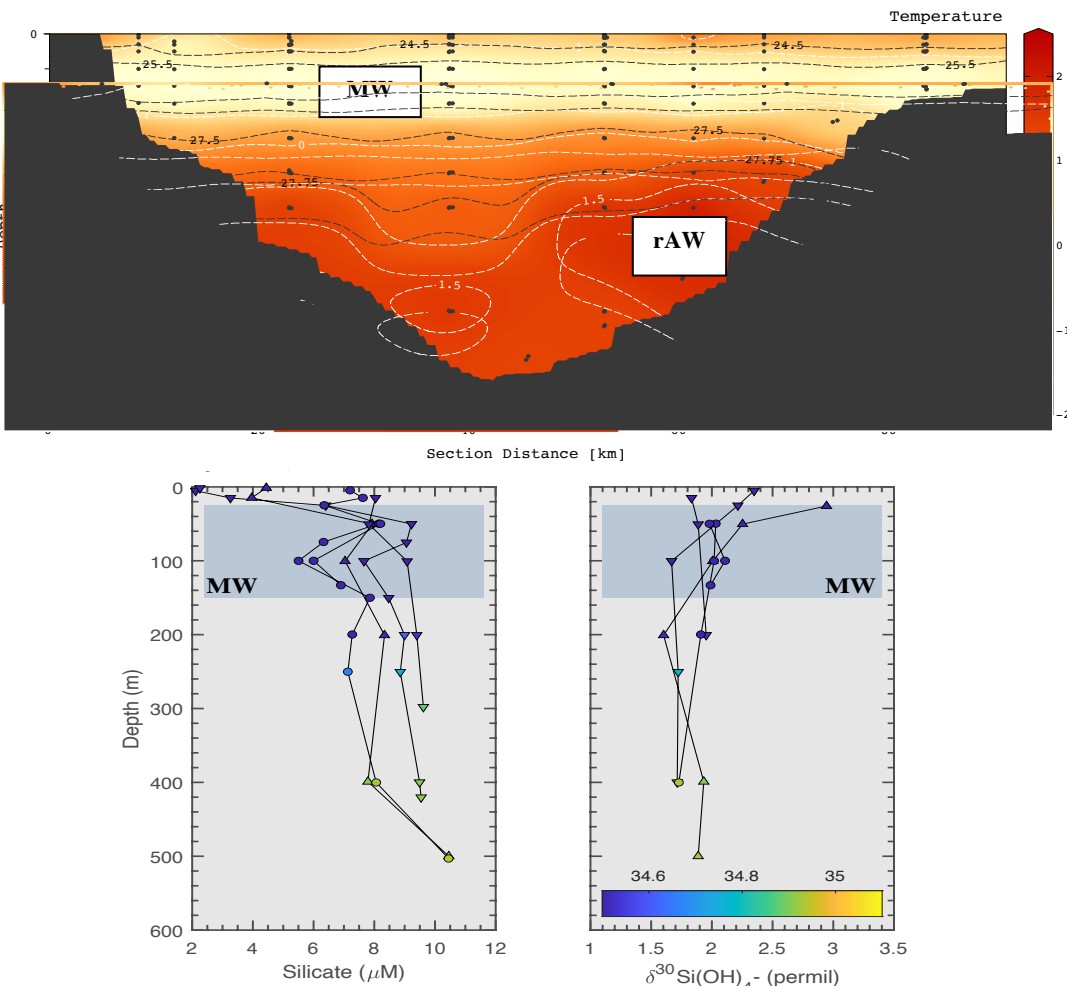

**Figure 9: Top: Integrated hydrography of the Ile-de-France section for 2017-2019. Isotherms are shown in white and isopycnals in black. MW = meteoric water, rAW = recirculated Atlantic Water. Bottom panels: DSi concentrations (left) and δ30Si(OH)4 (right) for late summer 2017 (circle), 2018 (upwards triangle) & 2019 (downwards triangle) of the Ile-de-France section. Colourscale represents salinity (psu).**

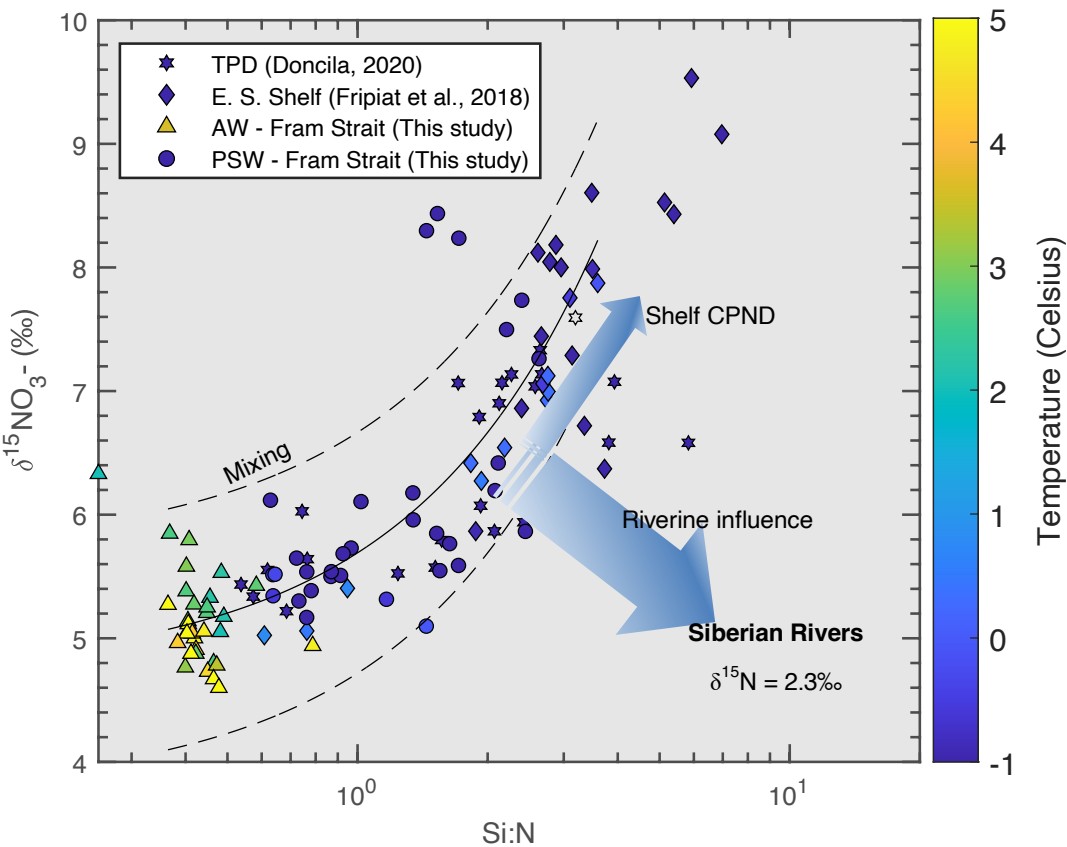

**Figure 10: Pan-Arctic trends of δ15N-NO3 against Si:N ratio. Triangles = Atlantic water at Fram Strait; Circles = Polar surface water at Fram Strait (this study). Stars = Transpolar drift (Doncila, 2020); Diamonds = East Siberian shelf (Fripiat et al., 2018). Dotted lines shows the regression line between AW and shelf endembers, dotted lines are for 1SD. Data is plotted below the mid-layer depth in Fram Strait to remove seasonal variation. This could not be applied to the transpolar drift and East Siberian shelf due to the shallowness of the watermasses. Colorscale shows temperature. δ15N-NO3 endmember for summertime Siberian rivers is obtained from Francis, 2019, from ArcticGRO measurements.**

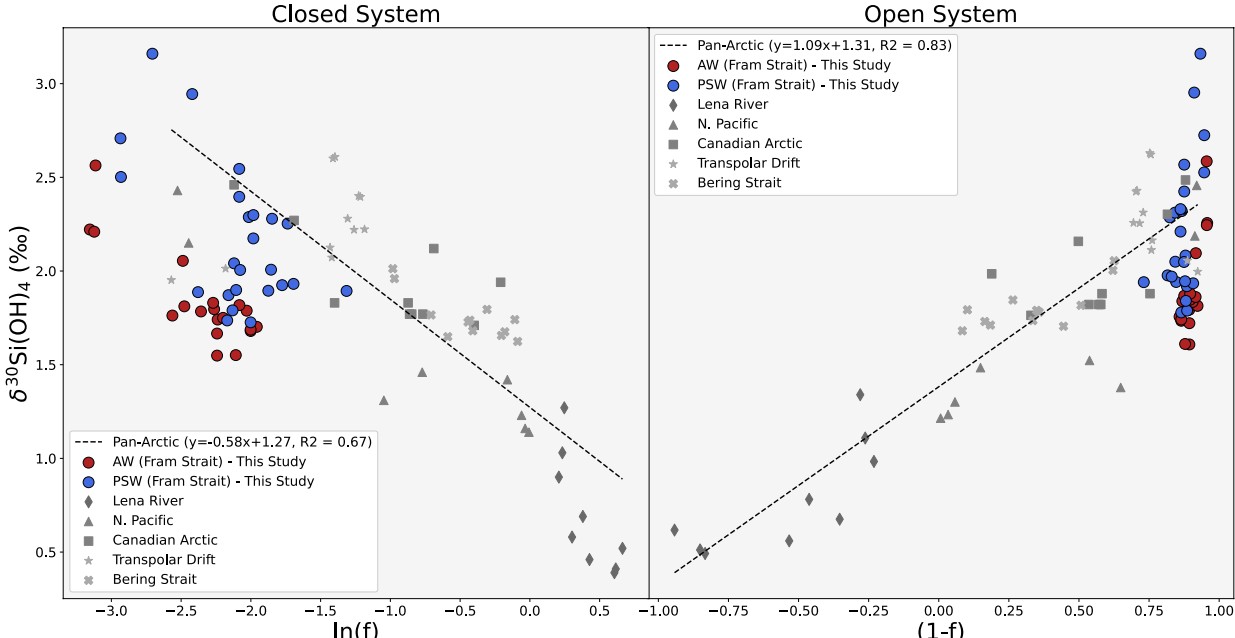

Figure 11: Estimate of the apparent $\delta^{30}$Si isotopic effect across the Arctic Ocean from source to export to Fram Strait through the transpolar drift. The lines and equations are the result of linear regression of $\delta^{30}$Si(OH)$_4$ vs the natural logarithm of f (where f = measured DSi/ DSi source prior to any biological consumption), representative of closed system dynamics (left) and $\delta^{30}$Si(OH)$_4$ vs 1-f, representative of open system dynamics. AW & PSW are not included in this regression as it is assumed they originate from different nutrient sources. Colored dots show measurements from within AW (red, max. depth = 600m) and PSW watermasses (blue, max.depth = 150m) from this study based on water mass definitions in Table 2. Grey symbol sets are published $\delta^{30}$Si(OH)$_4$ from major DSi sources to the surface Arctic domain and surface water masses. Triangles: N.Pacific (<100m, stations 1-6); Stars: Transpolar drift (<60m, stations 30-38 from Brzezinski et al., 2021). Crosses: Bering Strait (max. depth = 60m, stations 4-6 from Brzezinski et al., 2021). Squares: Canadian Arctic (surface and intermediate water mass signatures of the Canadian Arctic sector, from Table 2 in Giesbrecht et al., 2022). Octogones: Glacial runoff from Greenland and Svalbard glaciers (Hatton et al., 2019). Diamonds: Lena river (Sun et al., 2018). The y-intercept of both trendlines provide a $\delta^{30}$Si(OH)$_4$ estimate of DSi sources in the Arctic Ocean ranging from 1.27 – 1.31‰.

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
