# Peer review of "Tracing the role of Arctic shelf processes in Si and N cycling and export through the Fram Strait: Insights from combined silicon and nitrate isotopes"

_EGUsphere, 2022_

## Referee Comment (RC1)

This manuscript presents new nutrient and nutrient isotope data from across the Fram Strait, with the aim of deconvolving the controls on nitrogen and silicon cycling, and nitrate and dissolved silicon (DSi) biological uptake, in the Arctic. Given the rapid environmental (and biogeochemical) change in the Arctic, and the importance of this gateway for supplying nutrients to the North Atlantic, this paper is timely, the findings have key implications for our understanding of future change in the Arctic, and are of importance to the community. The manuscript is well-written and clearly presented, and an enjoyable read. As such, the manuscript is suitable for publication in an EGU journal. I just have a few suggestions for moderate revisions and broader discussions. I would also suggest the inclusion of some additional recent papers. These papers are only recently published (some only as pre-prints) so it's understandable that these could have been missed before submission, but I think it would be beneficial to include them now.

Most of my comments relate to the interpretation of the silicon isotope data, which have implications for the conclusions of the manuscript.

**1) One of the key challenges in this paper is in the interpretation of Si isotope systematics at very low DSi concentrations ([DSi]).** Specifically, I am concerned that without the very depleted DSi waters included in the isotopic analysis, it's challenging to determine the isotopic behaviour of DSi in the mixed layer (relevant for the discussion on line 269 onwards regarding the decoupling of N and Si[†]). From Figure 4, it's clear that there are several profiles missing between -5 and 5 degrees longitude. Furthermore, several very low [DSi] surface samples do not have corresponding $\delta^{30}Si(OH)_4$ measurements (which is entirely understandable as these are very challenging measurements to make!). It appears from Figure 4 that there is only one datapoint from within the mixed layer in the transect (although it's not clear how this corresponds to the data shown in Figure 6 – what depths are these data from?). So, interpreting the relationship between uptake/utilisation and isotopic composition (i.e., closed vs. open systematics) in the mixed layer is going to be challenging. As such, I would suggest perhaps toning down this aspect of the interpretation, relating to closed vs. open nature of DSi uptake. I was hoping to test out some of these plots myself, but I found that the data link was for the incorrect year of sampling, and the $\delta^{30}Si$ data were not available.

[†Note that if the discussion of N and Si decoupling is included, please note that this could be due to other related processes in addition to shifting limitation e.g., algal population structure changes throughout the season. If there are any data available on algal population structure, they would be useful to include or refer to here.]

Related to this question, I'm also intrigued as to why the observational data consistently fall 'below' the $\delta^{30}Si(OH)_4$ model curve in Figure 6 (c,d). It seems that, unlike for the N isotope model (Fig. 6 a,b), the model predicts heavier isotopic compositions in the seawater for DSi, regardless of assumptions about whether the system is open or closed. Why could this be? Is this related to an incorrect assumption about the endmember compositions, or related to recycling/dissolution?

**2) I don't think it's necessarily clear from the evidence presented that "DSi uptake is being regulated by nitrate availability" across the Arctic** (line 278, and very relevant for discussion on line 319 onwards). This could be the case in some regions of the Arctic, but I

think DSi uptake and biogenic silica formation will ultimately, on a Pan Arctic scale, be regulated by multiple factors. I think a lot of the nuance here lies in the different interpretations of limitation at different spatial and temporal scales, and different biological/ecological contexts. For example, the evidence noted by the authors for DSi limitation in summer seasons (especially in AW dominated surface waters) based on silicon-32 uptake experiments (e.g. Krause et al. papers) in addition to the relationship between [DSi] and nitrate concentrations. Giesbrecht et al., 2021, used similar experiments to show that Si uptake by diatoms was limited by DSi even high [DSi] availability in Pacific Arctic Region, likely because the diatoms were acclimated to high 'background' [DSi] conditions, and that there was also strong interaction between DSi and light limitation (also relevant for the discussion on line 466). In other words, Si uptake and biogenic silica formation can become limited before exhaustion of DSi in the water column, and can be promoted through addition of DSi even at relatively high background levels. Again, I would suggest that the authors rephrase these aspects of the paper to account for these complexities.

NB: Giesbrecht et al., 2022, also present $\delta^{30}Si(OH)_4$ data from the Pacific Arctic Region that needs to be included in this manuscript (and there is some very relevant content for the discussion on sea-ice on line 388 onwards).

**3) There could be more discussion surrounding the delivery of Si from glacial meltwater sources.** As mentioned on line 57, although there is a lot of both DSi and amorphous silica (ASi) in glacial meltwaters, we don't yet have a good handle on fjord and estuary processes (perhaps the authors could cite Meire et al., 2016, and Hopwood et al., 2020 here for discussion on this subject). I agree that the majority of studies show that surface waters within glacial fjords can be depleted to very low [DSi], and it's highly likely that much of the ASi will be trapped in fjord sediments (line 381 onwards). However, given the very high [ASi] in glacial meltwaters (an order of magnitude higher than [DSi], Hawkings et al., 2017) even a small percentage of ASi escaping the fjord system will contribute a significant amount towards the total silicon supply from glaciers. For example, a study of benthic Si cycling off SW Greenland revealed evidence for slow remineralization of isotopically light Si (a good proportion of which is likely to be glacial ASi) in coastal shelf sediments, driving a significant diffusive flux of DSi into overlying shelf waters (Ng et al., 2020). This ASi contribution would shift the glacial contribution in Figure 8 towards high ln(Si), with only a minor shift in $\delta^{30}Si$, closer towards the riverine endmember (but likely with an isotopically lighter composition). As such, this would make the glacial endmember more challenging to exclude. The questions of how much amorphous silica is present in Arctic river particulates, and how it impacts the overall isotopic composition of runoff, sadly remain unanswered.

**4) There is further information that could be included regarding the benthic supply of DSi to Arctic (shelf) waters** (relevant to discussion on e.g., line 307, 312, 320, 330). Ward et al., 2022, (GCA and Biogeosciences discussion) present new porewater/solid phase silicon isotope data from the Arctic. Given that these data were from samples that were collected in the Barents Sea, and so more relevant geographically than the study cited on line 211, I would suggest that this study is referenced here in addition to Ehlert et al., 2016. It might be possible (c. line 214), with the additional data and model results from Ward et al., to attempt to calculate possible fluxes of benthic DSi and their contribution to the overall water column signature (see their BGC discussion paper, Figure 6).

Further, these authors show that lithogenic silica (LSi) plays an important role in supplying DSi to Arctic shelf waters. Given that this LSi is isotopically light relative to river waters, this additional source exacerbates the Arctic isotope balance 'problem' (from Brzezinski et al., 2021). However, Ward et al. also discuss the possibility of abiotic *uptake* of Si into sediments (adsorption and – critically – authigenic phase formation) during early diagenesis that could form a sink of isotopically light Si. The longer-term BSi burial found by these authors is actually quite low due to strong benthic-pelagic coupling on a seasonal timescale.

**Minor points:**

Please add limits of detection and precision/accuracy data for the dissolved inorganic nutrient measurements section in the Methods (line 120-121). This is important when discussing such low nutrient concentrations, and using these low nutrient concentrations as an indication of ecosystem-scale nutrient limitation.

It's a shame that the $\delta^{30}$Si values were not able to be measured, as using the three isotopes is an extra check on data quality (through the determination of mass dependency). However, the $\delta^{29}$Si values of reference standards do seem to reproduce well and agree with published data. Please rephrase the sentence on line 146: whilst I agree it's a useful exercise for comparison with other datasets, I don't think that the conversion of $\delta^{29}$Si to $\delta^{30}$Si (by simply assuming mass dependency) results in better accuracy, *per se*.

Are there any seawater $\delta^{18}$O data that could be used together with salinity values to determine the contribution of meteoric and sea-ice sourced freshwater to the samples? This might help to deconvolve the runoff vs. sea ice contributions to the system.

It would be useful to add + or – signs before silicon isotope values for clarity.

Please be consistent with the use of high/low and heavy/light when referring to delta values or 'isotopic compositions/isotopic values/isotopically' (etc.) respectively.

I'm not sure that the authors need to include the dashed lines (showing the products of uptake e.g., biogenic silica) in Figure 6, given that solid phase data are not included in the paper. These could be removed for clarity.

Note that Lena is spelled incorrectly in the caption to Figure 8.

**References:**

Brzezinski, M. A., Closset, I., Jones, J. L., De Souza, G. F., & Maden, C. (2021). New constraints on the physical and biological controls on the silicon isotopic composition of the Arctic Ocean. Frontiers in Marine Science, 931.

Giesbrecht, K. E., & Varela, D. E. (2021). Summertime biogenic silica production and silicon limitation in the Pacific Arctic Region from 2006 to 2016. Global Biogeochemical Cycles, 35(1), e2020GB006629.

Giesbrecht, K. E., Varela, D. E., de Souza, G. F., & Maden, C. (2022). Natural variations in dissolved silicon isotopes across the Arctic Ocean from the Pacific to the Atlantic. Global Biogeochemical Cycles, 36(5), e2021GB007107.

Hawkings, J. R., Wadham, J. L., Benning, L. G., Hendry, K. R., Tranter, M., Tedstone, A., ... & Raiswell, R. (2017). Ice sheets as a missing source of silica to the polar oceans. Nature communications, 8(1), 1-10.

Hopwood, M. J., Carroll, D., Dunse, T., Hodson, A., Holding, J. M., Iriarte, J. L., ... & Meire, L. (2020). How does glacier discharge affect marine biogeochemistry and primary production in the Arctic?. The Cryosphere, 14(4), 1347-1383.

Meire, L., Meire, P., Struyf, E., Krawczyk, D. W., Arendt, K. E., Yde, J. C., ... & Meysman, F. J. R. (2016). High export of dissolved silica from the Greenland Ice Sheet. Geophysical Research Letters, 43(17), 9173-9182.

Ng, H. C., Cassarino, L., Pickering, R. A., Woodward, E. M. S., Hammond, S. J., & Hendry, K. R. (2020). Sediment efflux of silicon on the Greenland margin and implications for the marine silicon cycle. Earth and Planetary Science Letters, 529, 115877.

Ward, J. P., Hendry, K. R., Arndt, S., Faust, J. C., Freitas, F. S., Henley, S. F., ... & Tessin, A. C. (2022). Stable silicon isotopes uncover a mineralogical control on the benthic silicon cycle in the Arctic Barents Sea. Geochimica et Cosmochimica Acta.

Ward, J. P., Hendry, K. R., Arndt, S., Faust, J. C., Freitas, F. S., Henley, S. F., ... & Airs, R. L. (2022). Benthic Silicon Cycling in the Arctic Barents Sea: a Reaction-Transport Model Study. Biogeosciences Discussions, 1-34.

---

## Referee Comment (RC3)

**Review of manuscript egusphere-2022-254,**

In this manuscript, Debyser and co-workers present combined isotopic data of dissolved nitrate ($\delta^{15}$N-NO$_3$ and $\delta^{18}$O-NO$_3$) and silicon ($\delta^{30}$Si(OH)$_4$) in seawaters from the Fram Strait. By comparing the dataset between the inflowing Atlantic waters and the outflowing Pacific surface waters, the authors evaluate the modification of Si and N within the Arctic ocean and highlight the importance of denitrification on Arctic shelves as well as its impact on DSi utilization. The authors also try quantify the contribution of terrestrial input to the total DSi exported through Fram Strait.

This manuscript is well-constructed and well-written. This work improves the understanding of the modification of major nutrients (Si&N) within the Arctic Ocean, and has key implications for the future change of nutrient supply from the Arctic ocean to the Atlantic Ocean. As such, this work is a useful addition to the field that would be suitable for publication in BG. But there are a few points where I think the authors need to be more careful in their interpretation, as I outlined below. I recommend this manuscript for moderate to major revisions before the final publication.

**Major comments:**

1. The justification of measuring $\delta^{29}$Si instead of $\delta^{30}$Si

Method part, L145-L148: I understand that it is difficult to measure $\delta^{30}$Si of samples with low DSi concentrations. However, it is not common nowadays to report $\delta^{30}$Si values based on the measurements of $\delta^{29}$Si. This practice will be more justified if the authors could provide more details (can be in the supplementary) on why it was not possible to measure $\delta^{30}$Si directly. For example, what kind of efforts the authors have already put into trying to resolve the matrix effects and interferences? Normally anion doping (sulfate, nitrate, etc…) or pre-removal of organic matter could help to diminish the matrix effect (Closset et al., 2016; Hughes et al., 2011). Isobaric interferences of e.g., nitrogen ($^{14}$N$_2$) and nitric oxide ($^{14}$N$^{16}$O) can be avoided by medium resolution mode focusing on the left side of the peak shoulder (Liguori et al., 2020). At least it would be nice to see that the authors have already tried all these approaches before giving up on directly measuring $\delta^{30}$Si.

2. The interpretation of the isotopic fractionation model in section 4.1.1

   1) The initial level of the nutrients before utilization should be the subsurface water from the time period with strong mixing prior to the sampling, i.e. winter season. Initial condition for both spring and summer seasons should be the same (i.e. upwelled winter subsurface water). The authors did not give too much details of their choice on the initial condition, but it seems like they simply chose the subsurface waters in each sampling event as the initial condition. On the other hand, considering the horizontal transport of the water mass (i.e. PSW and AW), the initial condition might be found horizontally.

Therefore, I wonder whether the authors' choice of the initial condition of the model is correct and I recommend the authors to provide more information on this aspect.

2) It makes more sense to sperate PSW and AW dataset into different models because their initial conditions are different. Spring and summer dataset from the same water mass should be combined into the same model, because they belong to the same fractionation system, i.e. the nutrient kept being utilized in summer after the utilization in spring.

3) In figure 6, $\delta^{30}$Si data do not fit any of the models during any of the sampling events, so the discussion between L257 to L260 is not valid. This might point to the incorrect choice of initial condition that the authors applied to calculate the nutrient utilization.

3. The estimation of the terrestrial Si sources at Fram Strait in section 4.2.6

It is not justified to estimate the PSW Si:N (that free of terrestrial influence) based on the assumption of "further modification of marine $\delta^{30}$Si(OH)$_4$ and DSi:N through the Bering Strait is linear with North Pacific trends". Especially after the authors already concluded that nitrate was largely removed via denitrification in the Arctic, which will clearly modify the Si:N within the Arctic. It is thus not convincing to conclude DSi:N>0.78 in the PSW must originate from terrestrial riverine sources. Therefore, the estimation in the paragraph L434-441 is not valid. Additionally, as the authors illustrated the mixing scheme in Figure 10, it is quite obvious that Si:N of the PSW is located within the error of the mixing line between the AW and E.S. Shelf. This indicate that the increase of Si:N in the PSW is more a result of the shelf CPND. The only prominent outliers of PSW dataset beyond the mixing lines are the three data points with lower Si:N/heavier $\delta^{15}$N.

**Minor comments:**

L130-139: The two-step co-precipitation has been widely used previously, so there is no need to elaborate its necessity here. It can be directly cited from previous work, for example (Reynolds et al., 2006), (Grasse et al., 2013), (Liguori et al., 2020) etc…

L137: What does "regrouped" mean?

L152: Please note whether the uncertainties are 1SD or 2SD.

L181: Please add "in the upper 400m"

L185: Please add (Figure 3c)

L189: Please add (Figure 3b)

L193-195: Please tone down the argument here, as (5.42 ± 0.70 µM) and (6.65 ± 1.67 µM) are within error identical.

L196: Please add (Figure 3d)

Section 3.3: The description in this section is bouncing back and forth between figure 3 and figure 4&5, i.e. between whole depth profile and surface data. It will be clearer if the authors can give clearer information on which figure/panel the sentence refers to, and describe the distribution from the surface to the deep for both Si and N/O isotopes.

L234-235: "Nutrient utilization" is normally defined as the fraction of nutrient that has been utilized. The way that the authors define it here is against the common cognition.

L291-293: The whole sentence "settling particulate nitrogen… sediment interface." reads a bit repetitive, please rephrase.

Section 4.1.2 The authors try to discuss the modification of nitrate and DSi in the Arctic ocean in this section, so maybe the authors should exclude the dataset within the mixed layer, which are largely impacted by the local biological uptake. From Figure 7, only panel (c) excludes samples from within the mid-layer depth.

L355: It is not easy to understand "merging towards signatures resembling riverine endmembers" here, please give the values of the riverine endmembers.

L367: TDP → TPD

L372: The larger variability in Si isotope signatures of PSW ($R^2 > 0.3$) at Fram strait might reflect the combination of mixing and local biological uptake.

L376: valuated → evaluated

L397:

L409: ply?

Line 429-430: Please show the linear relationship between $\delta^{30}Si$ and DSi:N in North Pacific waters.

L489: I would not describe a 0.11‰ increase of $\delta^{30}Si(OH)_4$ as "significantly" enriched, as the long-term reproducibility of the ALOHA$_{1000m}$ measurement is 0.08‰, the two values with a difference of 0.11‰ even overlap within error.

Figures:

Figure 2: the scale of temperature (left panel) is missing

Figure 6: I wonder whether it is necessary to add the fractionation lines of the products, as there is no data from the biogenic phase and there's no discussion of the fractionation of the products. Removing these unnecessary lines can make the plots cleaner.

Figure 10: Please correct the sentence in the caption: " Solid line shows the regression (conservative mixing line) between AW and shelf  endmembers, dotted

lines are for one standard deviation." Also, if the line is conservative mixing line, then it is not regression line. They are not the same.

**References:**

Closset, I., Cardinal, D., Rembauville, M., Thil, F., & Blain, S. (2016). Unveiling the Si cycle using isotopes in an iron-fertilized zone of the Southern Ocean: From mixed-layer supply to export. *Biogeosciences*, *13*(21), 6049–6066. https://doi.org/10.5194/bg-13-6049-2016

Grasse, P., Ehlert, C., & Frank, M. (2013). The influence of water mass mixing on the dissolved Si isotope composition in the Eastern Equatorial Pacific. *Earth and Planetary Science Letters*, *380*, 60–71. https://doi.org/10.1016/j.epsl.2013.07.033

Hughes, H. J., Delvigne, C., Korntheuer, M., De Jong, J., André, L., & Cardinal, D. (2011). Controlling the mass bias introduced by anionic and organic matrices in silicon isotopic measurements by MC-ICP-MS. *Journal of Analytical Atomic Spectrometry*, *26*(9), 1892–1896. https://doi.org/10.1039/c1ja10110b

Liguori, B. T. P., Ehlert, C., & Pahnke, K. (2020). The Influence of Water Mass Mixing and Particle Dissolution on the Silicon Cycle in the Central Arctic Ocean. *Frontiers in Marine Science*, *7*(April), 1–16. https://doi.org/10.3389/fmars.2020.00202

Reynolds, B. C., Frank, M., & Halliday, A. N. (2006). Silicon isotope fractionation during nutrient utilization in the North Pacific. *Earth and Planetary Science Letters*, *244*(1–2), 431–443. https://doi.org/10.1016/j.epsl.2006.02.002

---

## Author Comment (AC1)

We thank Reviewer #1 for their time in reviewing our manuscript and their useful discussion concerning our findings. Here we include responses to all of the comments as follows: (1) Reviewer's comment (2) Author's comment (3) Suggested change to the manuscript.

(1) One of the key challenges in this paper is in the interpretation of Si isotope systematics at very low DSi concentrations ([DSi]). Specifically, I am concerned that without the very depleted DSi waters included in the isotopic analysis, it's challenging to determine the isotopic behaviour of DSi in the mixed layer (relevant for the discussion on line 269 onwards regarding the decoupling of N and Si[†]). From Figure 4, it's clear that there are several profiles missing between -5 and 5 degrees longitude. Furthermore, several very low [DSi] surface samples do not have corresponding $d^{30}Si(OH)_4$ measurements (which is entirely understandable as these are very challenging measurements to make!). It appears from Figure 4 that there is only one datapoint from within the mixed layer in the transect (although it's not clear how this corresponds to the data shown in Figure 6 – what depths are these data from?). So, interpreting the relationship between uptake/utilisation and isotopic composition (i.e., closed vs. open systematics) in the mixed layer is going to be challenging. As such, I would suggest perhaps toning down this aspect of the interpretation, relating to closed vs. open nature of DSi uptake. I was hoping to test out some of these plots myself, but I found that the data link was for the incorrect year of sampling, and the $d^{30}Si$ data were not available. [†Note that if the discussion of N and Si decoupling is included, please note that this could be due to other related processes in addition to shifting limitation e.g., algal population structure changes throughout the season. If there are any data available on algal population structure, they would be useful to include or refer to here.]

(2) Samples shown in Figure 4 are for FS2018 cruise only, while the dataset in Figure 6 includes both FS2018 and JR17005. The majority of the d30Si dataset from within the mixed layer is from JR17005. We have focused our measurements in the core of AW & PSW water masses ( -5˚ < & >5˚ longitude) as the physical circulation of central Fram Strait is strongly eddying and highly temporally variable, impeding straightforward interpretation of biogeochemical data. (3) The equivalent section of Figure 4 for JR17005 is now included in supplementary material S2. Samples from within the mixed layer are now highlighted in a different colour in Figure 6 for clarity. (2) As we have no measurements for either d15N &d30Si from within the mixed layer for later summer PSW as nutrient concentrations were depleted to nearly 0μM, (3) this aspect of the discussion has been toned down in the manuscript. The data link provided in the manuscript for biogeochemical data and d15N data of JR17005 have now been updated to the correct year.

(1) Related to this question, I'm also intrigued as to why the observational data consistently fall 'below' the $d^{30}Si(OH)_4$ model curve in Figure 6 (c,d). It seems that, unlike for the N isotope model (Fig. 6 a,b), the model predicts heavier isotopic compositions in the seawater for DSi, regardless of assumptions about whether the system is open or closed. Why could this be? Is this related to an incorrect assumption about the endmember compositions, or related to recycling/dissolution?

(2) Following suggestions from all reviewers, we have updated the parameters of our Rayleigh model in Figure 6 and the heavier prediction was related to incorrect assumptions

about endmember compositions. This is addressed in more details in the general comment to all reviewers.

(1) I don't think it's necessarily clear from the evidence presented that "DSi uptake is being regulated by nitrate availability" across the Arctic (line 278, and very relevant for discussion on line 319 onwards). This could be the case in some regions of the Arctic, but I think DSi uptake and biogenic silica formation will ultimately, on a Pan Arctic scale, be regulated by multiple factors. I think a lot of the nuance here lies in the different interpretations of limitation at different spatial and temporal scales, and different biological/ecological contexts. For example, the evidence noted by the authors for DSi limitation in summer seasons (especially in AW dominated surface waters) based on silicon- 32 uptake experiments (e.g. Krause et al. papers) in addition to the relationship between [DSi] and nitrate concentrations. Giesbrecht et al., 2021, used similar experiments to show that Si uptake by diatoms was limited by DSi even high [DSi] availability in Pacific Arctic Region, likely because the diatoms were acclimated to high 'background' [DSi] conditions, and that there was also strong interaction between DSi and light limitation (also relevant for the discussion on line 466). In other words, Si uptake and biogenic silica formation can become limited before exhaustion of DSi in the water column, and can be promoted through addition of DSi even at relatively high background levels. Again, I would suggest that the authors rephrase these aspects of the paper to account for these complexities.

(2) We agree with Reviewer #1 that nutrient regimes and subsequent limitation in the Arctic Ocean are strongly varying, and would like to emphasise that the conclusions drawn in this paper relating to N availability are relating to the Eurasian Arctic nutrient regime only, not to Arctic biological systems with high nutrient concentrations like ones from Pacific origins. (3) Certain aspects of the paper have been rephrased to clearly state that N-limitation and subsequent modulation of DSi uptake is observed in the Eurasian Arctic although we recognise that this does not apply to different nutrient regimes of the Arctic Ocean where biogeochemical cycles are vastly different. These aspects of the paper have been rephrased to account for the difference in regimes.

(1) Giesbrecht et al., 2022, also present $d^{30}Si(OH)_4$ data from the Pacific Arctic Region that needs to be included in this manuscript (and there is some very relevant content for the discussion on sea-ice on line 388 onwards). (3) Watermass endmember data from Giesbrecht et al. (2022) are now included in Figure 8.

(1) There could be more discussion surrounding the delivery of Si from glacial meltwater sources. As mentioned on line 57, although there is a lot of both DSi and amorphous silica (ASi) in glacial meltwaters, we don't yet have a good handle on fjord and estuary processes (perhaps the authors could cite Meire et al., 2016, and Hopwood et al., 2020 here for discussion on this subject). I agree that the majority of studies show that surface waters within glacial fjords can be depleted to very low [DSi], and it's highly likely that much of the ASi will be trapped in fjord sediments (line 381 onwards). However, given the very high [ASi] in glacial meltwaters (an order of magnitude higher than [DSi], Hawkings et al., 2017) even a small percentage of ASi escaping the fjord system will contribute a significant amount towards the total silicon supply from glaciers. For example, a study of benthic Si cycling off SW Greenland revealed evidence for slow remineralization of isotopically light Si (a good

proportion of which is likely to be glacial ASi) in coastal shelf sediments, driving a significant diffusive flux of DSi into overlying shelf waters (Ng et al., 2020). This ASi contribution would shift the glacial contribution in Figure 8 towards high ln(Si), with only a minor shift in d$^{30}$Si, closer towards the riverine endmember (but likely with an isotopically lighter composition). As such, this would make the glacial endmember more challenging to exclude. The questions of how much amorphous silica is present in Arctic river particulates, and how it impacts the overall isotopic composition of runoff, sadly remain unanswered.

(2) Thank you for this comment. We agree with the reviewer's comment that much remains to be understood regarding estuarine processes of glacial geochemical cycling and export to the ocean, and that it would be useful to understand the role of ASi in Arctic river particulates. (3) References to Hopwood et al. (2020) & Meire et al. (2016) have been added to improve the discussion from line 57 onwards regarding the characterization of the export of DSi and ASi into the open ocean. (2) We recognize that findings from Ng et al. (2020) show significant fluxes of DSi into shelf waters, however, it is hard to make this comparison to the data presented in Figure 8 without direct measurements from isotopic signatures from the shelf waters themselves and any discussion resulting from this would be speculative. As such, their research is now acknowledged in this paragraph but we refrain from making any direct comparison to this study.

(1) There is further information that could be included regarding the benthic supply of DSi to Arctic (shelf) waters (relevant to discussion on e.g., line 307, 312, 320, 330). Ward et al., 2022, (GCA and Biogeosciences discussion) present new porewater/solid phase silicon isotope data from the Arctic. Given that these data were from samples that were collected in the Barents Sea, and so more relevant geographically than the study cited on line 211, I would suggest that this study is referenced here in addition to Ehlert et al., 2016. It might be possible (c. line 214), with the additional data and model results from Ward et al., to attempt to calculate possible fluxes of benthic DSi and their contribution to the overall water column signature (see their BGC discussion paper, Figure 6). Further, these authors show that lithogenic silica (LSi) plays an important role in supplying DSi to Arctic shelf waters. Given that this LSi is isotopically light relative to river waters, this additional source exacerbates the Arctic isotope balance 'problem' (from Brzezinski et al., 2021). However, Ward et al. also discuss the possibility of abiotic *uptake* of Si into sediments (adsorption and – critically – authigenic phase formation) during early diagenesis that could form a sink of isotopically light Si. The longer-term BSi burial found by these authors is actually quite low due to strong benthic-pelagic coupling on a seasonal timescale.

(2) We thank the reviewer for bringing the two Ward manuscripts to our attention as we were not aware of their publishing at the time of writing our manuscript. They greatly add to our understanding of benthic efflux in the Arctic Ocean and (3) their measurements from the Barents Sea are now referred to on line 211 and included in the discussion from line 307 onward. (2) With regards to calculating fluxes of benthic DSi in this manuscript, the light d$^{30}$Si referred to from line 214 onwards are primarily observed in the deep basin of Fram Strait. Benthic cycling is vastly different in the Arctic Ocean on the shallow shelves compared to the deep basins (März et al., 2015), and we feel that as the model results from Ward et al. (2022) characterize benthic cycling on the shallow Barents sea shelf, these should not be directly applied to our measurements as they can be unrepresentative of the

biogeochemical cycling of the deep Fram Strait. As our measurements are also sparse in resolution above the sediment interface and not easily distinguished from advective signals, we maintain our opinion that our dataset precludes accurate quantification of recycling processes in the deep Fram Strait and we therefore refrain from doing so in this manuscript.

(1) Please add limits of detection and precision/accuracy data for the dissolved inorganic nutrient measurements section in the Methods (line 120-121). This is important when discussing such low nutrient concentrations, and using these low nutrient concentrations as an indication of ecosystem-scale nutrient limitation. (2) Detection limit for nutrient analysis was $0.1\mu M$ and $0.03\mu M$ for silicate and nitrate respectively with accuracy with respect to CRMS of 2.75% and 0.91% for JR17005. For FS2018, analytical precision is of 2% and the detection limit was of $0.4\mu M$ for nitrate and $0.1\mu M$ for silicate.  (3) Detection limit and precision data is included in method section.

(1) It's a shame that the $d^{30}Si$ values were not able to be measured, as using the three isotopes is an extra check on data quality (through the determination of mass dependency). However, the $d^{29}Si$ values of reference standards do seem to reproduce well and agree with published data. Please rephrase the sentence on line 146: whilst I agree it's a useful exercise for comparison with other datasets, I don't think that the conversion of $d^{29}Si$ to $d^{30}Si$ (by simply assuming mass dependency) results in better accuracy, *per se*.

(2) Following from another comment from Reviewer #3, we realise the writing of the method section read as if the d30Si isotope of samples was not measured during analysis and thank the reviewers for highlighting this. To clarify, all three isotopes were measured during analysis and a strong relationship was found between measured d29Si and d30Si (d29Si = 0.5131*d30Si, $R^2$ = 0.99), but larger variability and standard deviations were measured on the d30Si, hence the conversion from d29Si to d30Si. (3) The method description has now been amended for clarity on our method of analysis and the d29Si vs d30Si relationship of our measurements is now included in the supplementary material S1. (2) We agree with the comment on the conversion of both isotopes and (3) the sentence was edited to now read "$\delta^{29}Si$ were converted to $\delta^{30}Si$ to improve reliability and global comparability of datasets" instead.

(1) Are there any seawater $d^{18}O$ data that could be used together with salinity values to determine the contribution of meteoric and sea-ice sourced freshwater to the samples? This might help to deconvolve the runoff vs. sea ice contributions to the system. (2) We recognise that seawater $d^{18}O$ data could bring important hydrographical information, however, for the FS2018 cruise data was not available at the time of writing this manuscript, but we refer to the long term dataset from Dodd et al. (2012) to evaluate PSW composition.

(1) It would be useful to add + or – signs before silicon isotope values for clarity. (2) + / - signs were not originally added as all d30Si seawater measurements considered in this study were positive, (3) but these have now been added to silicon isotope values for clarity.

(1) Please be consistent with the use of high/low and heavy/light when referring to delta values or 'isotopic compositions/isotopic values/isotopically' (etc.) respectively. (3) Manuscript amended to use heavy/light only when referring to delta values for consistency

(1) I'm not sure that the authors need to include the dashed lines (showing the products of uptake e.g., biogenic silica) in Figure 6, given that solid phase data are not included in the paper. These could be removed for clarity. (3) The products line have been removed from Figure 6 in the main body of text for clarity, and solid phase data where available is now included in the supplementary section S3 alongside collection and method of analysis.

(1) Note that Lena is spelled incorrectly in the caption to Figure 8. (3) Spelling corrected in Figure 8 caption.

**References**

Dodd, P. A., Rabe, B., Hansen, E., Falck, E., MacKensen, A., Rohling, E., Stedmon, C. and Kristiansen, S.: The freshwater composition of the Fram Strait outflow derived from a decade of tracer measurements, J. Geophys. Res. Ocean., 117(11), 1–26, doi:10.1029/2012JC008011, 2012.

Giesbrecht, K. E., Varela, D. E., Souza, G. F. and Maden, C.: Natural Variations in Dissolved Silicon Isotopes Across the Arctic Ocean From the Pacific to the Atlantic, Global Biogeochem. Cycles, 36(5), 1–23, doi:10.1029/2021gb007107, 2022.

Hopwood, M. J., Carroll, D., Dunse, T., Hodson, A., Holding, J. M., Iriarte, J. L., Ribeiro, S., Achterberg, E. P., Cantoni, C., Carlson, D. F., Chierici, M., Clarke, J. S., Cozzi, S., Fransson, A., Juul-Pedersen, T., Winding, M. H. S. and Meire, L.: Review article: How does glacier discharge affect marine biogeochemistry and primary production in the Arctic?, Cryosphere, 14(4), 1347–1383, doi:10.5194/tc-14-1347-2020, 2020.

März, C., Meinhardt, A. K., Schnetger, B. and Brumsack, H. J.: Silica diagenesis and benthic fluxes in the Arctic Ocean, Mar. Chem., 171, 1–9, doi:10.1016/j.marchem.2015.02.003, 2015.

Meire, L., Meire, P., Struyf, E., Krawczyk, D. W., Arendt, K. E., Yde, J. C., Juul Pedersen, T., Hopwood, M. J., Rysgaard, S. and Meysman, F. J. R.: High export of dissolved silica from the Greenland Ice Sheet, Geophys. Res. Lett., 43(17), 9173–9182, doi:10.1002/2016GL070191, 2016.

Ng, H. C., Cassarino, L., Pickering, R. A., Woodward, E. M. S., Hammond, S. J. and Hendry, K. R.: Sediment efflux of silicon on the Greenland margin and implications for the marine silicon cycle, Earth Planet. Sci. Lett., 529, 115877, doi:10.1016/j.epsl.2019.115877, 2020.

Ward, J. P. J., Hendry, K. R., Arndt, S., Faust, J. C., Freitas, F. S., Henley, S. F., Krause, J. W., März, C., Ng, H. C., Pickering, R. A. and Tessin, A. C.: Stable Silicon Isotopes Uncover a Mineralogical Control on the Benthic Silicon Cycle in the Arctic Barents Sea, Geochim. Cosmochim. Acta, doi:10.1016/j.gca.2022.05.005, 2022.

---

## Author Comment (AC2)

We thank Prof Damien Cardinal for his time in completing this review. He has contributed to some very useful points to help improve this manuscript. Here we include responses to all of the comments as follows: (1) Reviewer's comment (2) Author's comment (3) Suggested change to the manuscript.

(1) Use dissolved silicon instead of silica: (3) Inconsistencies were addressed in the manuscript and "silica" replaced to "dissolved silicon" where relevant

(1) Fig. 1 and Table 1 and §2.1 From all these parts, it is unclear what are the sampling stations for N and Si isotopes from which cruises. There are 4 cruises on table 1, but in the paper isotopes data come mostly from only 2?... Please clarify.

(2) Thank you for highlighting this. The data presented in this manuscript for the main section across Fram Strait comes from two cruises (FS2018 & JR17005), with profiles for the Ile de France section from separate oceanographic cruises (FS2017, FS2018, FS2019, data presented in section 4.2.4 on sea-ice). (3) This has now been clarified in the Methods and Table 1 has been edited.

(1) In Fig. 1 mention that sea-ice extent displayed is from summer (I guess). There is no mention of N isotope sampling? (2) Sea ice extent is from September, (3) this has now been added to the Figure caption. (2) Nitrate isotope stations are shown by the blue & green dots on the map, silicon isotope stations are a subset of this. (3) This is now mentioned in the Figure 1 caption.

(1) L49 This sentence is unclear, since Pacific water are entering the Arctic, so the link with DSi export is not straightforward. Put "net supply of DSi" instead of "net export"? Or rephrase more clearly. (3) This sentence was rephrased to read "the excess of DSi in the Arctic Ocean's Si budget is attributed to Pacific water [...] and freshwater sources".

(1) L84 and 87-88, 93 provide st. dev. of these end-members isotopic signatures. (3) Stdvs for end-member isotopic signatures now included in the text.

(1) L86 Wrong reference. It should be Fripiat et al. 2018 instead of Fripiat et al. 2011 (3) Thank you for pointing out this mistake. It has now been edited to the correct reference

(1) L115: Samples have been deep-frozen, also for silicic acid concentration? This is not optimal since silica precipitation can take place during deep freezing. Please comment.

(2) We recognise that measurement from frozen is suboptimal for silicic acid concentrations, but separate non-frozen samples could not be collected for nutrients due to sampling and shipping restrictions. Concentrations were independently checked at the University of Edinburgh from the silicon isotope samples (acidified preserved) during analysis with the HACH reagent method. Both datasets were in very good agreement and frozen samples were not found to have lower DSi concentrations. DSi concentrations from FS2018 also closely align with concentrations measured in the same water masses in JR17005 below the seasonal layer in the upper 500m of the water column, and align with published concentrations in the literature. Through these combined methods of verification, we are confident on the validity on the accuracy of the concentrations measured by DTU.

(1) L120 Even though it is standard protocol, some reference on methods should be provided for nutrient analyses. (3) References to nutrient analysis of JR17005 (Brand et al., 2020) and FS2018 (Hansen and Koroleff, (1999) & Schnetger and Lehners, (2014)) are now included.

(1) L135 What is the difference between this preconcentration protocol and the one of Reynolds 2006, which has 2 steps too? (2) It follows the same principles as Reynolds et al. (2006) with increased sample volumes (40ml instead of 2ml) and reduced NaOH volumes for co-precipitation (1.1% v/v in $1^{st}$ step, 1% v/v in $2^{nd}$ step instead of 2%).

(1) Fig. 2 Legend is incomplete, e.g. there is no scale for the T°C panels. (3) Temperature scale added to figure 2 and legend has been updated.

(1) Table 2. Add st. dev. on all parameter water mass averages (e.g. NO3 and DSi concentrations, N*, Si*, capital delta). (3) Stdvs for all parameters now included in Table 2.

(1) Fig. 4 and 9. Be consistent with the name of the parameter, certainly no silica concentration is displayed here. Dissolved silicon or silicic acid would be much more appropriate. (3) Figure captions edited to read dissolved silicon instead of silica.

(1) L253-254. It is claimed here that AW follow more an open system with small fractionation. This is not so obvious from Fig. 6, especially for spring where almost no point fits the open model (grey line) except at low utilization (f close to 1) where both models cannot be differentiated. In summer, data are more consistent with open model, but then, why summer (i.e. more stratified I guess) would be more behaving as an open system? It would have been expected more from spring.

(2) Regarding the first part of this comment: Thank you for pointing this out, which we agree with. We have updated parameters of the Rayleigh model based on reviewers suggestions (see general comment to all reviewers for details of this). Based on the updated model, AW falls between closed and open conditions in AW, with a shift from more closed conditions in spring to more open conditions in summer. An increase in stratification would indeed lead to closed system fractionation. On the other hand, a shift from utilisation of "new" nitrate to regenerated nitrate is expected as nitrogen becomes depleted over the growth season and becomes more heavily recycled within the water column, leading the apparent fractionation trends to shift from closed to open system instead, as observed in our study.

(1) L262-263 The linearity between d30Si et DSi utilization is consistent with an isotopic fractionation highlighted L257, so, why say at the end of the § that it is mixing that control d30Si? Could mixing behavior be displayed on Fig. 6 to decipher? (2) This was a typo in the text and the sentence should have referred to the mixing behaviour of PSW instead. (3) This sentence has now been amended in the manuscript.

(1) Fig. 6 is there a justification having spring and summer displayed in different panels? Since they could a represent the same growth season / isotopic system, data could be merged? (2) Following suggestions from all reviewers we have updated our Rayleigh model and data is now separated by water masses instead to reflect different nutrient sources. This is further commented on in the general comment to all reviewers.

(1) Fig. 8. Mention from which depths the data have been taken. Is it only surface samples? Here also, the legend seems to be incomplete / erroneous. In the caption panel, the triangles are different from the graph (e.g. there are different colours, and different shapes with triangle tips up / down not consistent with the main graph). Consequently, I don't understand how the linear trendline for

AW has been drawn? How these AW and PSW trendlines compared with Rayleigh models displayed in Fig. 6?

(3) Figure 8 and its figure caption has now been updated for clarity (Figure 8 included below for reference). (2) The AW and PSW trendlines in Figure 8 follow observations from the Rayleigh models in Figure 6. Namely that DSi utilisation in AW follows closed system kinetics from isotopically light Atlantic DSi sources, while fractionation in PSW does not show a good fit with Rayleigh models. d30Si in PSW appears to be controlled by a mixture of AW and Arctic-sourced nutrients instead, plotting between the AW and Arctic trendlines.

[Figure]

**Figure 1: Pan-Arctic trends of $\delta^{30}$Si(OH)$_4$ against ln(DSi). Colored dots show measurements from within AW (red, max. depth = 600m) and PSW watermasses (blue, max.depth = 150m) from this study based on water mass definitions in Table 2. Grey symbol sets are published $\delta^{30}$Si(OH)$_4$ from major DSi sources to the surface Arctic domain and surface water masses. Triangles: N.Pacific (<100m, stations 1-6); Stars: Transpolar drift (<60m, stations 30-38 from Brzezinski et al., 2021). Crosses: Bering Strait (max. depth = 60m, stations 4-6 from Brzezinski et al., 2021). Squares: Canadian Arctic (surface and intermediate water mass signatures of the Canadian Arctic sector, from Table 2 in Giesbrecht et al., 2022). Octogones: Glacial runoff from Greenland and Svalbard glaciers (Hatton et al., 2019). Diamonds: Lena river (Sun et al., 2018). Stars show average endmember composition of AW (red) and Pacific and riverine sources (Grey). Red dotted trendline is the least-squared regression for $\delta^{30}$Si(OH)$_4$ vs the natural logarithm of DSi within AW, blue and grey dotted trendlines are the equivalent for PSW and pan-Arctic (excluding Fram Strait) respectively. These trendlines show fractionation from partial utilisation of DSi consistent with fractionation models.**

(1) L397 Weird wording here, probably "while" is not needed…(3) "While" removed from sentence.

(1) L407 who is Francis??? (2) This refers to data from the master's thesis by A. Francis (University of Edinburgh) on nitrate isotopes in Arctic rivers from ARCTICGRO samples (Francis, 2019). (3) This work is now directly referenced within the manuscript.

**References**

Brand, T., Norman, L., Mahaffey, C., Tuerena, R., Crocket, K. and Henley, S.: Dissolved nutrient samples collected in the Fram Strait as part of the Changing Arctic Ocean programme during cruise JR17005, May-June 2018., , doi:doi:10.5285/b61d58df-b8e8-11c4-e053-6c86abc0246c, 2020.

Francis, A.: Stable Isotope Tracing of Dissolved Nitrogen from Permafrost Degradation in Arctic Rivers, University of Edinburgh., 2019.

Hansen, H. P. and Koroleff, F.: Determination of nutrients, in Methods of Seawater Analysis, edited by K. Kremlingi and M. Ehrhardt, pp. 159–228, Verlag GmbH., 1999.

Reynolds, B. C., Frank, M. and Halliday, A. N.: Silicon isotope fractionation during nutrient utilization in the North Pacific, Earth Planet. Sci. Lett., 244(1–2), 431–443, doi:10.1016/j.epsl.2006.02.002, 2006.

Schnetger, B. and Lehners, C.: Determination of nitrate plus nitrite in small volume marine water samples using vanadium(III)chloride as a reduction agent, Mar. Chem., 160, 91–98, doi:10.1016/j.marchem.2014.01.010, 2014.

---

## Author Comment (AC3)

We thank Reviewer #3 for their time in reviewing our manuscript and the comments they have raised, particularly concerning the method of analysis, which have contributed to improving our work. Here we include responses to all of the comments as follows: (1) Reviewer's comment (2) Author's comment (3) Suggested change to the manuscript.

(1) Method part, L145-L148: I understand that it is difficult to measure $\delta^{30}$Si of samples with low DSi concentrations. However, it is not common nowadays to report $\delta^{30}$Si values based on the measurements of $\delta^{29}$Si. This practice will be more justified if the authors could provide more details (can be in the supplementary) on why it was not possible to measure $\delta^{30}$Si directly. For example, what kind of efforts the authors have already put into trying to resolve the matrix effects and interferences? Normally anion doping (sulfate, nitrate, etc...) or pre-removal of organic matter could help to diminish the matrix effect (Closset et al., 2016; Hughes et al., 2011). Isobaric interferences of e.g., nitrogen ($^{14}$N$_2$) and nitric oxide ($^{14}$N$^{16}$O) can be avoided by medium resolution mode focusing on the left side of the peak shoulder (Liguori et al., 2020). At least it would be nice to see that the authors have already tried all these approaches before giving up on directly measuring $\delta^{30}$Si.

(2) We thank reviewers #1 and #3 for their comments on measuring d29Si and for giving us the opportunity to elaborate on the methods of measurements, as we realise the writing of the method section reads as if the d30Si isotope of samples was not measured. To clarify, all three isotopes were measured during analysis and a strong relationship was found between measured d29Si and d30Si (d29Si = 0.5131*d30Si, R$^2$ = 0.99), but larger variability and standard deviations were measured on the d30Si. The left side of the peak shoulder was measured for each measurement in medium resolution as it is the standard practice of analysis for Nu plasma II MC-ICP-MS. An increasing number of laboratories has been reporting $\delta^{30}$Si values based on the measurements of $\delta^{29}$Si for the comparison of datasets, particularly at very low DSi concentrations (most recently for the Arctic Ocean: Liguori et al., 2021). Considering the robust relationship measured between d29Si and d30Si in our dataset, we found the method of conversion preferable to anion doping which we experienced within the wider context of this project and found to slightly increase variability in our measurements. (3) The method description has now been amended for clarity on our method of analysis and we will include a supplementary material section illustrating the d29Si vs d30Si of our measurements.

(1) The initial level of the nutrients before utilization should be the subsurface water from the time period with strong mixing prior to the sampling, i.e. winter season. Initial condition for both spring and summer seasons should be the same (i.e. upwelled winter subsurface water). The authors did not give too much details of their choice on the initial condition, but it seems like they simply chose the subsurface waters in each sampling event as the initial condition. On the other hand, considering the horizontal transport of the water mass (i.e. PSW and AW), the initial condition might be found horizontally. Therefore, I wonder whether the authors' choice of the initial condition of the model is correct and I recommend the authors to provide more information on this aspect. 2) It makes more sense to sperate PSW and AW dataset into different models because their initial conditions are different. Spring and summer dataset from the same water mass should be combined into the same model, because they belong to the same fractionation system, i.e. the nutrient kept being

utilized in summer after the utilization in spring. 3) In figure 6, $\delta^{30}$Si data do not fit any of the models during any of the sampling events, so the discussion between L257 to L260 is not valid. This might point to the incorrect choice of initial condition that the authors applied to calculate the nutrient utilization.

(2) We thank Reviewer #3 for their comment on the initial conditions of the Rayleigh fractionation model and general suggestions as we believe this has contributed to improve our manuscript. Following suggestions from all reviewers, we have updated the parameters of our Rayleigh model in Figure 6, improving the fit of the $\delta^{30}$Si data. (3) Panels are now separated by watermasses rather than by season. Initial conditions were chosen from subsurface waters and we discuss the importance of horizontal transport within the manuscript.(2) Adjustments made to the Rayleigh model is addressed in more details in the general comment to all reviewers.

(1) It is not justified to estimate the PSW Si:N (that free of terrestrial influence) based on the assumption of "further modification of marine $\delta^{30}$Si(OH)$_4$ and DSi:N through the Bering Strait is linear with North Pacific trends". Especially after the authors already concluded that nitrate was largely removed via denitrification in the Arctic, which will clearly modify the Si:N within the Arctic. It is thus not convincing to conclude DSi:N>0.78 in the PSW must originate from terrestrial riverine sources. Therefore, the estimation in the paragraph L434-441 is not valid.

(2) We have taken this comment onboard and have now independently evaluated the composition of PSW DSi using the estimated Arctic-wide fractionation factor from the robust pan-Arctic Rayleigh trend (R$^2$= 0.67 & 0.83) shown in Figure 8 (now including data from Giesbrecht et al., 2022) & subsequent apparent utilisation of nutrients in PSW. Closed and open fractionation trends are shown in the two graphs below (included in supplementary material S4). Using this method, we calculate a pan-Arctic isotope effect 30e,

and calculate that around 50% of nutrients from shelf and halocline waters are utilised in PSW. From this updated calculation we find riverine sources contribute to 40 ± 4% of the total DSi inventory at Fram Strait, with Pacific sources contribute to around 8 ± 1%. We believe this new calculation to be more robust than the original evaluation based on Si:N ratio and (3) the discussion in the paragraph L434-441 has been updated accordingly.

(1) Additionally, as the authors illustrated the mixing scheme in Figure 10, it is quite obvious that Si:N of the PSW is located within the error of the mixing line between the AW and E.S. Shelf. This indicate that the increase of Si:N in the PSW is more a result of the shelf CPND. The only prominent outliers of PSW dataset beyond the mixing lines are the three data points with lower Si:N/heavier δ15N.

(2) As per our discussion in section 4.1.2, we agree that the dominant influence on d15N is from shelf CPND, which will impact Si:N. Terrestrial influence is however observed on the Pan-Arctic trend, and to a smaller extent, within PSW, with lower d15N and higher Si:N than expected from the linear regression. Although the PSW dataset fall within 1SD of the linear regression, a large part of the measurements fall below the mean where DSi:N>1. This trend is further supported by seperate measurements from the Eurasian shelves (Laptev Sea) and over the continental slope (Debyser, in preparation).

(1) L130-139: The two-step co-precipitation has been widely used previously, so there is no need to elaborate its necessity here. It can be directly cited from previous work, for example (Reynolds et al., 2006), (Grasse et al., 2013), (Liguori et al., 2020) etc... (3) Method has now been edited down.

(1) L137: What does "regrouped" mean?  (2) This refers to the regrouping of the two separate precipitate during the two-step preconcentration.

(1) L152: Please note whether the uncertainties are 1SD or 2SD. (2) Uncertainties are 1SD. (3) This is now included in the text.

(1) L181: Please add "in the upper 400m". (3) Added

(1) L185: Please add (Figure 3c).  (3) Added

(1) L189: Please add (Figure 3b). (3) Added

(1) L193-195: Please tone down the argument here, as (5.42 ± 0.70 µM) and (6.65 ± 1.67 µM) are within error identical.  (3) Sentence now edited to read  "Below the mixed layer, DSi is low in AW (5.42 ± 0.70 µM) from DSi poor Atlantic waters of sub-tropical origins. DSi in PSW is slightly higher albeit within error of AW (6.65 ± 1.67 µM), potentially reflecting Arctic sources of DSi to PSW."

(1) L196: Please add (Figure 3d). (3) Added

(1) Section 3.3: The description in this section is bouncing back and forth between figure 3 and figure 4&5, i.e. between whole depth profile and surface data. It will be clearer if the authors can give clearer information on which figure/panel the sentence refers to, and

describe the distribution from the surface to the deep for both Si and N/O isotopes. (3) References to panels and figures for each sentence of section 3.3 are now included.

(1) L234-235: "Nutrient utilization" is normally defined as the fraction of nutrient that has been utilized. The way that the authors define it here is against the common cognition. (3) Nutrient utilization changed to nutrient fraction (f) instead in the manuscript and figure captions.

(1) L291-293: The whole sentence "settling particulate nitrogen... sediment interface." reads a bit repetitive, please rephrase. (3) Sentence rephrased to read "[…] settling particulate organic nitrogen from coastal productivity degrades at the sediment interface of the extensive shallow shelves and produces large sources of sedimentary ammonium."

(1) Section 4.1.2 The authors try to discuss the modification of nitrate and DSi in the Arctic ocean in this section, so maybe the authors should exclude the dataset within the mixed layer, which are largely impacted by the local biological uptake. From Figure 7, only panel (c) excludes samples from within the mid-layer depth.

(2) We believe it is important to include datasets within the mixed layer in Figure 7.a, b and d as they illustrate how the variation in nutrients in both water masses lead to diverging trends within the surface layer, linking remote nutrient modification to biological trends at Fram Strait.

(1) L355: It is not easy to understand "merging towards signatures resembling riverine endmembers" here, please give the values of the riverine endmembers. (3) Values of riverine endmembers added to L355.

(1) L367: TDP → TPD. (3) Typo amended

(1) L372: The larger variability in Si isotope signatures of PSW ($R^2>0.3$) at Fram strait might reflect the combination of mixing and local biological uptake. (2) We agree with this, (3) sentence has now been rephrased to read "[…] reflects the combined effects of local biological uptake and mixing between Arctic and Atlantic source signatures around Fram Strait."

(1) L376: valuated → evaluated. (3) Typo amended

(1) L397: while. (3) "While" removed from sentence.

(1) L409: ply? (3) Sentence rephrased to "[…] by mixing across the halocline in basins where AAW underlies below PSW" for clarity.

(1) Line 429-430: Please show the linear relationship between $\delta^{30}Si$ and DSi:N in North Pacific waters. (2) Following our answer to comment 3 above, this is no longer relevant to

the discussion as estimations are no longer based on DSi:N and this has been removed from the discussion.

(1) L489: I would not describe a 0.11‰ increase of $\delta^{30}$Si(OH)$_4$ as "significantly" enriched, as the long- term reproducibility of the ALOHA$_{1000m}$ measurement is 0.08‰, the two values with a difference of 0.11‰ even overlap within error. (2) The enrichment we measure align with trends of enrichment measured across the Arctic (Varela et al., 2016, Brzezinski et al., 2021, Giesbrecht et al., 2022). We are confident that this is a trend within our dataset and not measurement error and have decided to keep the mention of this trend in our conclusions.

(1) Figure 2: the scale of temperature (left panel) is missing. (3) Figure is amended and temperature scale is now included.

(1) Figure 6: I wonder whether it is necessary to add the fractionation lines of the products, as there is no data from the biogenic phase and there's no discussion of the fractionation of the products. Removing these unnecessary lines can make the plots cleaner. (3) Product fractionation lines have been removed from Figure 6 , and biogenic phase measurements are now included in this work in the supplementary material S3.

(1) Figure 10: Please correct the sentence in the caption: "Dotted lines Solid line shows the regression (conservative mixing line) between AW and shelf endembers endmembers, dotted lines are for one standard deviation." Also, if the line is conservative mixing line, then it is not regression line. They are not the same. (2) The line displayed in Figure 10 is the regression line, not a conservative mixing line and (3) the caption has been edited to "Solid line shows the linear regression between AW and shelf endembers, dotted lines are for one standard deviation".

**References**

Giesbrecht, K. E., Varela, D. E., Souza, G. F. and Maden, C.: Natural Variations in Dissolved Silicon Isotopes Across the Arctic Ocean From the Pacific to the Atlantic, Global Biogeochem. Cycles, 36(5), 1–23, doi:10.1029/2021gb007107, 2022.

Liguori, B. T. P., Ehlert, C., Nöthig, E. M., van Ooijen, J. C. and Pahnke, K.: The Transpolar Drift Influence on the Arctic Ocean Silicon Cycle, J. Geophys. Res. Ocean., 126(11), doi:10.1029/2021JC017352, 2021.

---

## Author Comment (AC4)

**Reply to all reviewers regarding comments on the isotopic fractionation model (section 4.1.1)**

We thank all three reviewers for their judicious comments and suggestions on this section as we believe it has helped bring clarity to this section and improve the manuscript overall.

Following comments from Reviewers #2 and #3, PSW and AW datasets are now divided into separate models to reflect the different nutrient sources for initial conditions. Building on the comments raised by Reviewer #1, the colour code of the fractionation model now clearly highlights which samples are from within the mixed layer for clarity of interpretation of data at low nutrient concentrations. Updated Figure 6 is included below for reference.

Initial conditions prior to utilization were chosen from the subsurface water nutrient concentrations from the water mass (Table 2) as data on winter nutrient concentrations in Fram Strait is limited, particularly for PSW. The study from Randelhoff et al. (2018) shows that summer nutrient concentrations below the mixed layer depth are a good approximation of winter concentrations. The authors found using these conditions preferable to finding horizontal conditions instead due to the large uncertainty arising from the spatial and temporal variability expected for these conditions. This is further discussed in the edited manuscript. Adjusted initial conditions address the issue of AW d30Si measurements artificially falling below the predicted compositions raised by Reviewers #1 and #3, now reflecting the lower DSi nature of AW. PSW does not fit either model of fractionation, which we attribute to the lateral transport of DSi signal instead. The updated model has not majorly changed observed trends but greatly improved fit of d30Si data and has provided clarity for interpretation. The discussion of all trends relating to Figure 6 have been updated accordingly.

[Figure]

**Figure 1: Top panels: nitrate utilisation vs $\delta^{15}$N-NO$_3$ for AW (left) and PSW (right). Bottom panels: DSi utilisation vs $\delta^{30}$Si(OH)$_4$ for AW (left) and PSW (right). Circles are denote measurements from JR17005 (spring) and triangles from FS2018 (summer). Red symbols show measurements within the mixed layer. Black line follows the closed fractionation model and grey line an open fractionation model.**

**References**

Randelhoff, A., Reigstad, M., Chierici, M., Sundfjord, A., Ivanov, V., Cape, M., Vernet, M., Tremblay, J.-É., Bratbak, G. and Kristiansen, S.: Seasonality of the Physical and Biogeochemical Hydrography in the Inflow to the Arctic Ocean Through Fram Strait, Front. Mar. Sci., 5(June), 1–16, doi:10.3389/fmars.2018.00224, 2018.

---

## Referee Report (RR1)

This manuscript is a revised version of egusphere-2022-254 that I also reviewed. Dr. Debyser and coauthors have put a great deal of work into improving the manuscript, including (a) modifying the fractionation models and (b) changing the approach to evaluate the composition of PSW DSi.

In general, the revisions have significantly strengthened the manuscript, and my recommendation is to

accept after my two concerns are addressed as detailed below, including

1) some clarification in the method part, and 2) more rigorous statistical analyses in section 4.1.1

1) Method part on the Si isotopes measurement:

The authors should clarify, if they produced duplicate and triplicate measurements and mention that in the method part. It is very uncommon for stable Si isotopes in seawater if all the samples were only measured ones (using a standard-bracketing method with 3-4 replicates) as they are normally not volume limited. It is very important to have at least parts of the samples measured as full replicates including the MAGIC precipitation step and column chemistry and run in different analytical sessions (eg. De Souza et al., 2012; Grasse et al., 2013; Liguori et al., 2021). This is especially important in this study, as 1) $\delta^{30}Si$ in the study underlies many different processes, 2) samples with low DSi concentrations are difficult to measure correctly due to high matrix/Si ratio, 3) The authors did not measure $\delta^{30}DSi$ directly, but instead only measured $\delta^{29}Si$.

And also, please report the uncertainties as 2SD, which is a common practice in reporting seawater Si isotopes data, and which will be useful for lab comparison.

2) Statistical analyses for discussions in section 4.1.1

The authors stated "AW follows the traditional isotopic effect of 5‰ and PSW follows the particularly low isotopic effect of 2‰", but it is not obvious in Figure 6. If the authors plot fractionation curves with $\varepsilon$=2‰ on the left upper plot, they should also fit the dataset as much as the curves with $\varepsilon$=5‰. The authors should do statistical analyses for example curve fitting to support this statement.

Same goes to the statement "In PSW… A shift towards a mostly linear trend in summer is observed, suggesting open system kinetics below the mixed layer", it is difficult to see this trend from the right upper plot in Fig.6. There are also some triangle datapoints visually follow the exponential trend. So please use statistical analyses to support this statement. This is very important for the follow up discussion to be convincing.

Minor comments:

1) The authors should also be careful with some terms/concepts that are discussed in the manuscript, for example: "$\delta^{30}Si(OH)_4$ in PSW does not show a good fit with either of the Rayleigh models"

Not both models are called Rayleigh model. One is called a "Rayleigh model", and the other is called a "open system/steady state model"

2) In the paragraph Line 458 to 463, there's a logic gap from previous discussion to "this suggests that….PSW carries a remote signal of partial DSi uptake, and is controlled by mixing and dilutive effects rather than local biological processes". In fact, this is a conclusion to draw after the discussion in section 4.2. So, it came a bit too early to place this conclusion already in section 4.1.1, or more discussions/arguments are needed here.

References:

De Souza, G.F., Reynolds, B.C., Rickli, J., Frank, M., Saito, M.A., Gerringa, L.J.A., Bourdon, B., 2012. Southern Ocean control of silicon stable isotope distribution in the deep Atlantic Ocean. Global Biogeochem. Cycles 26, 1–13. https://doi.org/10.1029/2011GB004141

Grasse, P., Ehlert, C., Frank, M., 2013. The influence of water mass mixing on the dissolved Si isotope composition in the Eastern Equatorial Pacific. Earth Planet. Sci. Lett. 380, 60–71. https://doi.org/10.1016/j.epsl.2013.07.033

Liguori, B.T.P., Ehlert, C., Nöthig, E.M., van Ooijen, J.C., Pahnke, K., 2021. The Transpolar Drift Influence on the Arctic Ocean Silicon Cycle. J. Geophys. Res. Ocean. 126. https://doi.org/10.1029/2021JC017352

---

## Referee Report (RR2)

This work is very interesting and worth for publication in BG. The revised version has significantly improved the manuscript with appropriate reply to most of my comments.
I still have few minor/moderate suggestions listed below.

Damien Cardinal

L 100-103, L260 and all over the manuscript. The st dev provided in the revised version were needed are welcome. It is however not obvious that isotopic signatures are significantly different (e.g. 5.5 +/- 0.4 vs. 5.1 +/- 0.2 pmil). Simple stats should be reported to limit the discussion to significant differences only. Note that this remark applies to the whole manuscript, often – but not always – average and SD are provided, but never p-value and significance of the differences when comparing concentration or isotopic signatures of water masses (t-test is probably appropriate most of the time).

L124-125 Unclear / meaningless. Need to rephrase.

L150-168 It'd be good to put the answer to my comment on frozen samples and DSi measurements in the revised paper to inform the readers that this has been handled and also to underline that it's not a standard protocol.

Supplementary material. I did not find any of the 4 supp. mat., which prevented me to evaluate them and the discussion on the main text where they are referred…

Fig. 6 Incomplete caption. To which depths these lines were drawn (it's mentioned that red dots are from the mixed layer, but what is the depth range of the black data?). Refer in the caption to how f has been calculated (table 2).

Fig. 8 + L460 + L 615 + section 4.2.6 starting L715: A fig. d30Si vs 1/DSi would really help to look at mixing. It could ideally be in a second panel in Fig. 8 since the current panel (and the text) clearly shows that PSW are largely scattered and are not explained by the Rayleigh system displayed.

Section 4.2.6 and Fig. S4. This figure is very informative, I'd put it in the main manuscript. With this figure, there could be a brief discussion on what model is the more likely to represent the data. It is quite clear that the open model is more coherent since (i) r2 is much better than Rayleigh and, (ii) the slope is more consistent with global epsilon (-1pmil).

---

## Author Response (AR2)

We thank Prof Damien Cardinal and Reviewer #3 for their additional comments on our manuscript. Here we include responses to all of the comments as follows: (1) Reviewer's comment **(2) Author's comment** *(3) Suggested change to the manuscript.*

**Response to Reviewer #3**

(1) The authors should clarify, if they produced duplicate and triplicate measurements and mention that in the method part. It is very uncommon for stable Si isotopes in seawater if all the samples were only measured once (using a standard-bracketing method with 3-4 replicates) as they are normally not volume limited. It is very important to have at least parts of the samples measured as full replicates including the MAGIC precipitation step and column chemistry and run in different analytical sessions (eg. De Souza et al., 2012; Grasse et al., 2013; Liguori et al., 2021). This is especially important in this study, as 1) $\delta^{30}$Si in the study underlies many different processes, 2) samples with low DSi concentrations are difficult to measure correctly due to high matrix/Si ratio. **(2) With the addition of the international seawater standard ALOHA for seawater silicon isotope measurements, it is not uncommon to not run full method replicates on each individual seawater samples (Giesbrecht et al., 2022; Ng et al., 2020; Cao et al., 2012). This is because running several ALOHA standards also provides full replicates of the full preparation method and analytical procedure over the entire measurement period. In the case of this study, those very good intra-run reproducibility where 1SD = 0.08‰ and 0.05‰ for ALOHA$_{1000}$ and ALOHA$_{300}$ across 58 and 30 individual measurements respectively, including full method repeats. The reproducibility of the full chemical and analytical procedure, which includes chemical preparation and isotopic measurements in separate analytical sessions, was additionally estimated on a subset of duplicate samples ($n$ = 8). The mean absolute difference between duplicate samples analyzed in this way was 0.04‰ (1 SD).** *(3) The above sentence is now included on L223.*

(1) 3) The authors did not measure $\delta^{30}$DSi directly, but instead only measured $\delta^{29}$Si. **(2) As already stated in the supplementary material and explained in the previous round of reviews, $\delta^{30}$DSi was also measured, but we have chosen to convert $\delta^{29}$Si to $\delta^{30}$Si as a precaution against any interferences that could have affected $\delta^{30}$Si in low concentration samples.**

(1) Please report the uncertainties as 2SD, which is a common practice in reporting seawater Si isotopes data, and which will be useful for lab comparison. **(2) Uncertainties for Si isotopes remain commonly reported in 1SD in seawater, modern and past work (Brzezinski & Jones, 2015; Closset et al, 2022; Dumont et al, 2020). Thus for consistency with N isotope datasets and as it does not impede comparison with other datasets, we have kept our measurements reported to 1SD.**

(1) The authors stated "AW follows the traditional isotopic effect of 5‰ and PSW follows the particularly low isotopic effect of 2‰", but it is not obvious in Figure 6. If the authors plot fractionation curves with ε=2‰ on the left upper plot, they should also fit the dataset as much as the curves with ε=5‰. The authors should do statistical analyses for example curve fitting to support this statement. Same goes to the statement "In PSW... A shift towards a mostly linear trend in summer is observed, suggesting open system kinetics below the mixed layer", it is difficult to see this trend from the right upper plot in Fig.6. There are also some triangle datapoints visually follow the exponential trend. So please use statistical analyses to support this statement. This is very important for the follow up discussion to be convincing. **(2) This discussion refers to N isotopic trends previously described for the published dataset in Tuerena et al., 2021, with justification and statistical analysis therein.** *(3) "As described in Tuerena et al. (2021a) for this dataset" added to L384 and reference to the same paper on L628.*

(1) The authors should also be careful with some terms/concepts that are discussed in the manuscript, for example: "$\delta^{30}Si(OH)_4$ in PSW does not show a good fit with either of the Rayleigh models". *(3) Sentence on L530 was changed to read "$\delta^{30}Si(OH)_4$ in PSW does not show a good fit with either of the fractionation models"*

(1) In the paragraph Line 458 to 463, there's a logic gap from previous discussion to "this suggests that....PSW carries a remote signal of partial DSi uptake, and is controlled by mixing and dilutive effects rather than local biological processes". In fact, this is a conclusion to draw after the discussion in section 4.2. So, it came a bit too early to place this conclusion already in section 4.1.1, or more discussions/arguments are needed here. *(3) L533-534 rephrased to read "This suggests that unlike nitrate, DSi in PSW is not primarily controlled by biological processes, and its variations are more likely to be driven by physical mixing and dilutive effects instead." instead, following the logic of line 530-533 above.*

**References**

Brzezinski, Mark A., and Janice L. Jones. 2015. "Coupling of the Distribution of Silicon Isotopes to the Meridional Overturning Circulation of the North Atlantic Ocean." *Deep-Sea Research Part II: Topical Studies in Oceanography* 116: 79–88. https://doi.org/10.1016/j.dsr2.2014.11.015.

Cao, Zhimian, Martin Frank, Minhan Dai, Patricia Grasse, and Claudia Ehlert. 2012. "Silicon Isotope Constraints on Sources and Utilization of Silicic Acid in the Northern South China Sea." *Geochimica et Cosmochimica Acta* 97: 88–104. https://doi.org/10.1016/j.gca.2012.08.039.

Closset, Ivia, Mark A. Brzezinski, Damien Cardinal, Arnaud Dapoigny, Janice L. Jones, and Rebecca S. Robinson. 2022. "A Silicon Isotopic Perspective on the Contribution of Diagenesis to the Sedimentary Silicon Budget in the Southern Ocean." *Geochimica et Cosmochimica Acta* 327: 298–313. https://doi.org/10.1016/j.gca.2022.04.010.

Dumont, M., L. Pichevin, W. Geibert, X. Crosta, E. Michel, S. Moreton, K. Dobby, and R. Ganeshram. 2020. "The Nature of Deep Overturning and Reconfigurations of the Silicon Cycle across the Last Deglaciation." *Nature Communications* 11 (1): 1–11. https://doi.org/10.1038/s41467-020-15101-6.

Giesbrecht, Karina E., Diana E. Varela, Gregory F. Souza, and Colin Maden. 2022. "Natural Variations in Dissolved Silicon Isotopes Across the Arctic Ocean From the Pacific to the Atlantic." *Global Biogeochemical Cycles* 36 (5): 1–23. https://doi.org/10.1029/2021gb007107.

Ng, Hong Chin, Lucie Cassarino, Rebecca A. Pickering, E. Malcolm S. Woodward, Samantha J. Hammond, and Katharine R. Hendry. 2020. "Sediment Efflux of Silicon on the Greenland Margin and Implications for the Marine Silicon Cycle." *Earth and Planetary Science Letters* 529: 115877. https://doi.org/10.1016/j.epsl.2019.115877.

Tuerena, R.E., J. Hopkins, P. J. Buchanan, R. S. Ganeshram, L. Norman, W-Jvon-Appen, A. Tagliabue, et al. 2021. "An Arctic Strait of Two Halves: The Changing Dynamics of Nutrient Uptake and Limitation across the Fram Strait." *Global Biogeochemical Cycles*. https://doi.org/10.1029/2021gb006961.

**Response to Damien Cardinal**

(1) L 100-103, L260 and all over the manuscript. The st dev provided in the revised version were needed are welcome. It is however not obvious that isotopic signatures are significantly different (e.g. 5.5 +/- 0.4 vs. 5.1 +/- 0.2 p: mil). Simple stats should be reported to limit the discussion to significant differences only. Note that this remark applies to the whole manuscript, often – but not always – average and SD are provided, but never p-value and significance of the differences when comparing concentration or isotopic signatures of water masses (t-test is probably appropriate most of the time). *(3) t-test values were added to L268, L278, L332, L346-347.*

(1) L124-125 Unclear / meaningless. Need to rephrase. *(3) L124-125 We rephrased this statement to "This heavy isotopic enrichment is attributed to physical processes (Liguori et al., 2020) and biological modification within surface waters (Giesbrecht et al., 2022; Varela et al., 2016)."*

(1) L150-168 It'd be good to put the answer to my comment on frozen samples and DSi measurements in the revised paper to inform the readers that this has been handled and also to underline that it's not a standard protocol. *(3) Added to L167 "While measurement from frozen is suboptimal for silicic acid concentrations, separate non-frozen samples could not be collected for nutrients due to sampling and shipping restrictions. DSi concentrations were independently checked at the University of Edinburgh from the silicon isotope samples (acid preserved) during analysis with the HACH reagent method. Both datasets from frozen and acidified were in very good agreement and frozen samples were not found to have lower DSi concentrations. DSi concentrations from FS2018 also closely align with concentrations measured in the same water masses in JR17005 below the seasonal layer in the upper 500m of the water column, and align with published concentrations in the literature."*

(1) Fig. 6 Incomplete caption. To which depths these lines were drawn (it's mentioned that red dots are from the mixed layer, but what is the depth range of the black data?). Refer in the caption to how f has been calculated (table 2). *(3) Depth range of the black data plotted is now included in the caption of Fig.6: "Nitrate utilisation vs $\delta^{15}N\text{-}NO_3$ for AW (left, depth <600m) and PSW (right, depth < 150m). Bottom panels: DSi utilisation vs $\delta^{30}Si(OH)_4$ for AW (left, depth <600m) and PSW (right, depth < 150m)." (3) Also added to figure caption: "f is the fraction of nutrient remaining, calculated from the nutrient concentrations of water masses AW and PSW below the MLD (Table 2)."*

(1) Fig. 8 + L460 + L 615 + section 4.2.6 starting L715: A fig. d30Si vs 1/DSi would really help to look at mixing. It could ideally be in a second panel in Fig. 8 since the current panel (and the text) clearly shows that PSW are largely scattered and are not explained by the Rayleigh system displayed.

**(2) While we agree that d30Si vs 1/DSi can be a useful way to display data and look at mixing, in the case of looking at a wide area over the Arctic Ocean, where both mixing and uptake come into play and the range of DSi concentrations is large, we do not believe that displaying the data in this way significantly improves the discussion from section 4.2.6 onwards, as shown on the plot below. As such, it was not included in the manuscript.**

[Figure]

(1) Section 4.2.6 and Fig. S4. This figure is very informative, I'd put it in the main manuscript. With this figure, there could be a brief discussion on what model is the more likely to represent the data. It is quite clear that the open model is more coherent since (i) r2 is much better than Rayleigh and, (ii) the slope is more consistent with global epsilon (-1pmil). *(3) Figure S.4 is now included in the main text as Figure 11, with brief discussion on the fit of models from L809-813.*